# Understanding Chain-of-Thought in LLMs through Information Theory

## Abstract

Large Language Models (LLMs) have shown impressive performance in complex reasoning tasks through the use of Chain-of-Thought (CoT) reasoning, allowing models to break down problems into manageable sub-tasks. However, existing CoT evaluation techniques either require annotated CoT data or fall short in accurately assessing intermediate reasoning steps, leading to high rates of false positives. In this paper, we formalize CoT reasoning in LLMs through an information-theoretic lens. Specifically, our framework quantifies the 'information gain' at each reasoning step, enabling the identification of failure modes in LLMs without the need for expensive annotated datasets. We demonstrate the efficacy of our approach through extensive experiments on toy and GSM-8K data, where it significantly outperforms existing outcome-based methods by providing more accurate insights into model performance on individual tasks.

## 1 Introduction

Large Language Models (LLMs) have demonstrated remarkable capabilities across a wide range of tasks, from complex reasoning to code generation (Chowdhery et al., 2024; OpenAI et al., 2024; Bubeck et al., 2023; Anil et al., 2023). Many of these advances can be attributed to Chain-of-Thought (CoT) reasoning (Wei et al., 2024; Nye et al., 2021; Li et al., 2024), which involves breaking down complex problems into a series of intermediate steps, mirroring human-like reasoning processes. The success of CoT reasoning, particularly in domains such as mathematics, logic, and multi-step decision-making, has led researchers and developers to incorporate CoT-like features directly into model training, i.e. the FLAN family of models (Chung et al., 2022; Wei et al., 2022).

This paper introduces a new formal framework for analyzing CoT in LLMs. We provide a rigorous method grounded in information theory, to evaluate the quality of each step in a model's reasoning process, thus offering insights beyond simple accuracy metrics to identify areas for improvement.

Previous work in this area has proposed "*Process Supervision*" (Lightman et al., 2023), which requires expensive, human-annotated step-by-step data. While effective, this approach is often impractical due to the high cost and effort of creating large-scale annotated datasets. In turn, alternative methods have recently been proposed, such as outcome reward modelling (Havrilla et al., 2024) or the Math-Shepherd (Wang et al., 2024b). Both these approaches avoid reliance on annotated step-wise CoT data by instead modelling the correctness of each step based on the correctness of final outputs. However, as we demonstrate in this paper, these methods can be unsound for detecting incorrect reasoning steps and can thus lead to a high false-positive rate in certain scenarios.

To address these shortcomings, we employ an information-theoretic approach, grounded in the following key insight: *Each correct step in a reasoning process should provide valuable and relevant information that aids in predicting the final correct outcome*. Building on this insight, we develop a framework to quantify the "*information gain*" after each sub-task in the reasoning process, without the need for step-by-step annotations. This enables us to detect sub-tasks that fail to contribute meaningful information toward the correct solution, signalling potential errors or irrelevant steps in the model's reasoning. In addition, we also introduce a practical algorithm to assess LLM performance across various sub-tasks within a Chain-of-Thought (CoT) reasoning process.

The key contributions of this paper are as follows:

1. We develop a framework for sequential applications of sub-tasks, e.g. Chain-of-Thought and provide a rigorous language to describe and detect detect failure modes in LLMs.

2. Based on this framework, we propose a practical algorithm to assess the task-wise performance of models. This yields more granular information about a model's CoT performance without requiring annotated data for intermediate reasoning steps.

3. We validate our methods on extensive toy data and the GSM-8K dataset (Cobbe et al., 2021). Our method effectively identifies failure modes in CoT reasoning, unlike baselines like outcome reward modelling (Havrilla et al., 2024) and Math-Shepherd (Wang et al., 2024b), which rely on final accuracy and tend to increase false positives in error detection.

## 2 PROPOSED FRAMEWORK: SETUP AND NOTATION

Before diving into our framework, we first provide a high-level overview and notation on how LLM generation will be treated throughout this paper. This will allow us to set the foundation for describing our information-theoretic framework. In particular, following the approach in González & Nori (2023), we view LLMs as abstract execution machines with a natural language interface. From this perspective, prompts are designed to solve specific problems (e.g., mathematical or logical problems), and the LLM processes the information in the prompt to generate an output.

We now define the notation for a typical prompt as a combination of two components:

1. An initial state, represented by a random variable $X_0 \in \mathcal{X}$, denotes information provided in the prompt that the LLM must operate on to obtain the queried information.

2. A task $\lambda \in \Upsilon$ (e.g., addition followed by multiplication) which encapsulates how the LLM should process information in $X_0$.

Given the prompt, defined as a tuple $(X_0, \lambda)$, the state $X_1$ represents the result of applying task $\lambda$ to the initial state $X_0$. Formally, we denote this using the *update* mapping $\Lambda : \mathcal{X} \times \Upsilon \to \mathcal{X}$ which outputs the updated state $X_1$ by applying the task $\lambda$ on $X_0$, i.e. $X_1 = \Lambda(X_0, \lambda)$. This updated state is then used to obtain the final output, denoted by $Y \in \mathcal{X}$, by extracting only the information in $X_1$ which is relevant to the queried final answer. This notation defines a prompt that instructs a model to process information drawn from some initial distribution $p(X_0)$ (e.g., mathematical problems).

Let us use the following simple example to illustrate the notation:

**Prompt:** *"James has 3 apples and Abbey has 9. How many apples do the two have in total?"*

Here, using the above notation, the initial state $x_0$ denotes the information *"James has 3 apples; Abbey has 9 apples"*, and $\lambda$ denotes the addition task. Next, $x_1 = \Lambda(x_0, \lambda)$ represents the updated information after correctly performing the addition operation, i.e. $x_1 =$*"James has 3 apples; Abbey has 9 apples; The two have 12 apples in total"*. The final output, $y$, is then obtained by simply extracting the total number of apples from $x_1$, i.e. *"The two have 12 apples in total"*[1]. With this basic notation established, we now consider compositions of tasks, enabling us to formalize the Chain of Thought (CoT) process in LLMs.

### 2.1 COMPOSITIONALITY

Many mathematical or logical problems require a sequential application of operations. Our notation is also amenable to such problems as it accommodates the composition of tasks. Consider a problem which requires two successive steps to arrive at the correct output:

$$\textbf{Prompt: } \textit{"Solve for } z = 2 \times (x + y) \textit{ where } x = 12 \textit{ and } y = 13\textit{"}. \tag{1}$$

In this example, first, we apply the addition operation to find the value of $x + y$, and next, we apply the multiplication operation to find the value of $z$. Using our notation this can be expressed as $\Lambda(x_0, \lambda_1 \circ \lambda_2)$, where $\lambda_1, \lambda_2$ denote the addition and multiplication tasks respectively. The following property allows us to concretely define the application of compositional task $\lambda_1 \circ \lambda_2$:

---

[1]Our setup also encapsulates cases with ambiguous (or multiple correct) responses for a given task $\lambda$ and initial state $x_0$. In this case, $\Lambda(x_0, \lambda)$ is a random variable with distribution $p(X_1 \mid X_0 = x_0)$. Therefore, for generality, we treat $\Lambda(x_0, \lambda)$ as a random variable from now on.

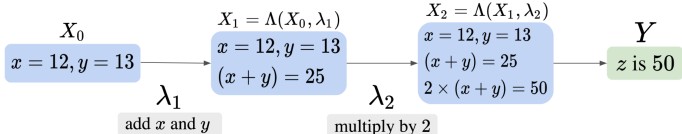

Figure 1: Solving the problem in prompt (1) requires compositional application of tasks.

**Definition 2.1.** We say that an update rule $\Lambda : \mathcal{X} \times \Upsilon \to \mathcal{X}$ is *compositionally consistent* if:

$$\Lambda(x_0, \lambda_1 \circ \lambda_2) \stackrel{\mathrm{d}}{=} \Lambda(\Lambda(x_0, \lambda_1), \lambda_2) \qquad \text{for all } x_0 \in \mathcal{X} \text{ and } \lambda_1, \lambda_2 \in \Upsilon.$$

Here, $\stackrel{\mathrm{d}}{=}$ denotes equality in distribution and is sufficient in many cases. For example, where a query may have multiple correct responses, an almost sure equality may be too restrictive.

Going back to the prompt in (1), Figure 1 shows that the model first computes $x + y$, and next multiplies the result by 2. Here, we refer to $X_1, X_2$ as *intermediate* states and $Y$ is the correct final output. More generally, if a problem statement requires sequential application of $T$ sub-tasks, $\lambda = \lambda_1 \circ \ldots \circ \lambda_T$, then the Chain-of-Thought (CoT) reasoning is divided up into $T$ steps, where the output of the $t$'th step is recursively defined as $X_t = \Lambda(X_{t-1}, \lambda_t)$ for $t \in \{1, \ldots, T\}$. Finally, the overall true output $Y$ is obtained by extracting the queried information from the final state $X_T$.

Having established a formal language for the sequential application of tasks, e.g. CoT, we now turn towards how a task may be divided into such a sequence of intermediate sub-tasks.

## 2.2 PRIMITIVE TASKS

In this subsection, we introduce the notion of *primitive tasks* which form the basic building blocks of any task. Intuitively, our formulation is reminiscent of ideas from linear algebra, where basis vectors form the basic building blocks of a vector space. In our case, any task $\lambda \in \Upsilon$ can be expressed as a sequence of primitive tasks. This decomposition will allow us to establish which tasks the model could have learned from the training data. For example, if a specific primitive task is not available in the LLM training data, it would be impossible for the model to execute any instructions which involve this primitive task correctly. With this in mind, we now introduce this concept formally:

**Definition 2.2** (Primitive tasks). We say that a set of tasks $\Gamma \subseteq \Upsilon$ is primitive if, for any task $\lambda \in \Upsilon$, there exists a unique subset $\{\lambda_i\}_{i=1}^k \subseteq \Gamma$ such that $\lambda = \lambda_1 \circ \cdots \circ \lambda_k$.

Note that the decomposition is not unique but the set of components is. In some cases, there may exist distinct permutations of primitive tasks which compose to yield the same task as is common in many associative operations. As an example, in the context of mathematical problem-solving, the basic arithmetic operation could be considered primitive. The composition of these primitive tasks allows us to construct extremely complex operations. Just like in linear algebra, we define the span of these tasks as the set obtained by their sequential applications.

**Definition 2.3** (Span of tasks). Let $\Phi \subseteq \Upsilon$ be a set of tasks, then:

$$\mathrm{Span}(\Phi) = \{\lambda_1 \circ \ldots \circ \lambda_k : \lambda_i \in \Phi \text{ for } 1 \leq i \leq k, k \in \mathbb{Z}_{>0}\}.$$

The set $\mathrm{Span}(\Phi)$ comprises all the tasks that can be applied by composing sub-tasks in the set $\Phi$. This means that any *compositionally consistent* update rule $\Lambda$ which is well-defined on the set of tasks $\Phi$ will also be well-defined on $\mathrm{Span}(\Phi)$. However, this $\Lambda$ may still be ill-defined for any task not in this span. This limitation is captured by the concept of unidentifiability, which plays a central role in determining the boundaries of what a model can and cannot infer.

## 2.3 UNIDENTIFIABILITY

The unidentifiability of tasks forms a key part of our framework. It directly addresses the fundamental challenge that models, such as LLMs, face when dealing with unseen tasks. If a task $\lambda$ lies outside of $\mathrm{Span}(\Phi)$, the span of tasks the model has been trained on, then the model cannot be expected to infer or apply it correctly. In other words, the model's capacity is constrained by the identifiability of tasks within the training set. This notion and formalization of unidentifiability allows us to highlight

a critical limitation in the generalization of models: tasks not encountered during training cannot be reliably executed, as they remain beyond the model's learned task span. More formally:

**Definition 2.4** (Unidentifiability). Let $\Phi \subseteq \Upsilon$ be any set of tasks, then a tasks $\lambda$ is said to be unidentifiable in $\Phi$ iff, $\lambda \notin \text{Span}(\Phi)$.

**Remark** In practice, the concept of unidentifiability may depend on the initial state $X_0$. For instance, an LLM might accurately perform addition for 2-digit numbers but fail with 10-digit numbers (Razeghi et al., 2022). Our framework can be extended to account for such cases by explicitly incorporating the distribution of initial states into the notion of identifiability. For example, addition could be considered unidentifiable when the initial state distribution is $p(X_0 \mid X_0$ includes 10-digit numbers). However, for simplicity, we keep this distributional dependence implicit in the definition provided earlier.

With this general framework in place, we can now turn this theoretical foundation into a practical algorithm for detecting unidentifiable sub-tasks. Specifically, we explore how the notion of unidentifiability can be combined with information-theoretic approaches to detect failure points in LLMs.

## 3 Operationalising our framework

This section aims to operationalise the above framework to make inferences regarding the unidentifiability of intermediate sub-tasks in a model's CoT reasoning process. This would subsequently allow us to detect any sub-task at which a model's CoT reasoning process starts to diverge from the ground truth, thereby providing insights into how the model can be improved. For example, suppose we are in a setting where the "addition" operation is unidentifiable, then we could further improve the model's mathematical reasoning by fine-tuning it on the addition operation.

### 3.1 An information-theoretic perspective

To make the concept of unidentifiability practical in the context of CoT generations, we begin by introducing the fundamental assumption. The core assumption in our approach is that each correctly executed CoT reasoning step should contribute meaningful and relevant information that aids in predicting the correct final output, denoted as $Y$. If we encounter a step after which the amount of information regarding $Y$ stops increasing, then we can take this as an indication of an incorrectly executed task. We concretise this assumption using using our notation from the previous section:

**Assumption 3.1** (Bayesian network). Let $\lambda \neq \lambda'$ be two operations with primitive decompositions:

$$\lambda = \lambda_1 \circ \ldots \lambda_{k-1} \circ \lambda_k \circ \cdots \circ \lambda_T \quad \text{and} \quad \lambda' = \lambda_1 \circ \ldots \lambda_{k-1} \circ \lambda'_k \circ \cdots \circ \lambda'_{T'},$$

where $\lambda'_k$ is unidentifiable in $\{\lambda_1, \ldots, \lambda_T\}$. Then, the intermediate states corresponding to the tasks $\lambda, \lambda'$ have the following Bayesian network:

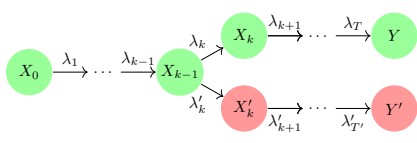

Figure 2: Bayesian network

**Intuition** The Bayesian network in Figure 2 implies that for any two reasoning paths which diverge at step $k$, the future states $X_i$ and $X'_j$ for any $i, j \geq k$ satisfy the conditional independence $X_i \perp\!\!\!\perp X'_j \mid X_{k-1}$. Consequently, once we apply $\lambda'_k$, the subsequent states along the new reasoning path (in red) add no information regarding the subsequent states or the output of the original path (in green). Hence the figure represents the fact that, for any given input, the output of $\lambda_k$ (top fork) contains no information regarding the output of any other primitive task $\lambda'_k$ (bottom fork).

Now that we have formalised our key information-theoretic assumption on the ground-truth CoT process, we turn towards the model behaviour on unidentifiable tasks in the following section.

### 3.2 Task execution in LLMs

To operationalise our framework, we formally distinguish between the model i.e. LLM's task execution and the *ground truth* process which arises from following the instructions correctly. To this end, we explicitly define how an LLM interprets a specified task $\lambda$ using the update rule, $\Lambda^M(X_0, \lambda)$, which is in general distinct from the *ground truth* update rule $\Lambda(X_0, \lambda)$.

Here, one option would be to consider the idealised setting where the model learns to perfectly follow some of the primitive tasks available in the training data. However, this may be considered too restrictive since in reality most LLMs do not always follow a "learned" task perfectly. Instead, we consider a much weaker assumption that the model cannot correctly execute a task which is unidentifiable in the training data. To this end, suppose $\Gamma^M \subseteq \Gamma$ denotes the primitive tasks available in the LLM training data. Concretely, we make the following assumption on LLM's task execution.

**Assumption 3.2** (Task execution in LLMs). $\Lambda^M$ is compositionally consistent and for any $(x_0, \lambda) \in \mathcal{X} \times \Upsilon$, there exists some $\widehat{\lambda} \in \mathrm{Span}(\Gamma^M)$ such that $\Lambda^M(x_0, \lambda) \stackrel{\mathrm{d}}{=} \Lambda(x_0, \widehat{\lambda})$.

**Intuition** Assumption 3.2 means that for any task which we would like the LLM to apply, the LLM ends up executing some task in $\mathrm{Span}(\Gamma^M)$ which the model has been trained on. In other words, the model's execution is restricted only to the tasks which could be inferred from the training data (i.e. in $\mathrm{Span}(\Gamma^M)$). Moreover, this assumption also allows us to encapsulate cases where the model does not follow the correct instructions or does not decompose a given task correctly.

Before proceeding further with our main result which will allow us to test for the unidentifiability of sub-tasks, we define some notation which we will use from now onwards. Let $\lambda = \lambda_1 \circ \ldots \circ \lambda_T$ denote a primitive decomposition of a task $\lambda$. Then, starting from an initial state $X_0$, we denote the model's intermediate states recursively as:

$$X_t^M := \Lambda^M(X_{t-1}^M, \lambda_t) \quad \text{and} \quad X_0^M = X_0.$$

Moreover, we use $Y^M$ to denote the model's final output. Next, using this notation, we present the conditional independence which must hold if the model encounters an unidentifiable intermediate task along its CoT reasoning path.

**Theorem 3.3.** *Let $\Gamma^M \subseteq \Gamma$ denote the primitive tasks available in the training data. Let $\lambda$ be a task with decomposition $\lambda = \lambda_1 \circ \ldots \circ \lambda_T$. If $\lambda_k$ is the first task in the decomposition of $\lambda$ which is unidentifiable in $\Gamma^M$ (i.e. $k = \arg\min_t\{\lambda_t \notin \mathrm{Span}(\Gamma^M)\}$). Then, under Assumptions 3.1 and 3.2, we have that*

$$Y \perp\!\!\!\perp X_j^M \mid X_{j-1}^M \quad \text{for all } j \geq k. \tag{2}$$

Theorem 3.3 shows that under Assumptions 3.1 and 3.2, when the model encounters an unidentifiable task (i.e. $\lambda_k$ in Theorem 3.3) in its Chain-of-Thought reasoning, the model output satisfies the conditional independence in Equation (2). More concretely, after a model's CoT reasoning diverges from the ground truth at step $k$, every subsequent step adds no additional information regarding the correct final output $Y$. In practice, this 'information' is measured by checking if the model's confidence about the final output $Y$ increases after each step. This is formalised in the next section.

### 3.3 TESTING FOR UNIDENTIFIABILITY USING INFORMATION GAIN

Having established all the essential components of our framework, we can now provide a concrete description of how to practically identify unidentifiable sub-tasks using information theory. As is common in the literature (Wang et al., 2024b; Havrilla et al., 2024), we assume access to a dataset consisting of prompts and their corresponding final answers, obtained by correctly applying the task $\lambda$. This dataset is denoted as $\mathcal{D}_\lambda := \{(x_0^i, y^i)\}_{i=1}^n$.

Additionally, recall that $X_j^M$ and $X_{j-1}^M$ represent the model's chain of thought (CoT) reasoning at steps $j$ and $j-1$, respectively. Consequently, each element in the conditional independence statement in Equation (2) can be derived from the data and/or the model.

To this end, we consider the mutual information between $Y$ and $X_j^M$ conditional on $X_{j-1}^M$, denoted by $\mathcal{I}(Y; X_j^M \mid X_{j-1}^M)$. This conditional mutual information term intuitively represents the *additional information* contributed by the $j$'th step of CoT, that is relevant for predicting the ground truth final output $Y$. Therefore, we refer to $\mathcal{I}(Y; X_j^M \mid X_{j-1}^M)$ as the *information gain* at step $j$.

It follows from Theorem 3.3 that if an LLM encounters a sub-task at step $i$ which is unidentifiable in its training data, no subsequent step should contribute any additional information relevant for predicting $Y$ (i.e. the information gain should remain 0 after step $i$). If, on the other hand, we observe that $\mathcal{I}\left(Y; X_j^M \mid X_{j-1}^M\right) > 0$ for some $j \geq i$, then under Assumptions 3.1 and 3.2, the task $\lambda_i$ is not unidentifiable. To estimate the information gain in practice, we use the following result:

**Proposition 3.4.** *Let $\mathcal{I}(X; Y \mid Z)$ denote the mutual information between random variables $X$ and $Y$ conditional on $Z$. Then,*

$$\mathbb{E}[\log p(Y \mid X_j^M)] - \mathbb{E}[\log p(Y \mid X_{j-1}^M)] = \mathcal{I}\left(Y; X_j^M \mid X_{j-1}^M\right) \geq 0. \tag{3}$$

To estimate the information gain in (3) using Proposition 3.4, we train a separate LLM, which we refer to as the *supervisor model* $g_{\text{sup}}$. This model takes as input the model's CoT reasoning up to any given intermediate step $t$, $X_t^M$, and is fine-tuned to directly predict the ground truth final output $Y$. In this way $g_{\text{sup}}(X_t^M)$ approximates the conditional distribution $p(Y \mid X_t^M)$. Then, the quantity $\mathbb{E}[\log p(Y \mid X_j^M)]$ can be estimated using the negative cross-entropy loss for predicting $Y$, i.e.

$$\mathbb{E}[\log p(Y \mid X_j^M)] \approx \mathbb{E}[\log \hat{p}(Y \mid X_j^M)] = -\mathbb{E}[l_{\text{CE}}\left(Y, g_{\text{sup}}(X_j^M)\right)],$$

where $l_{\text{CE}}$ denotes the cross-entropy loss. From this, it follows that

$$\underbrace{\mathbb{E}[\log p(Y \mid X_j^M)] - \mathbb{E}[\log p(Y \mid X_{j-1}^M)]}_{\text{Information gain}} \approx \mathbb{E}[l_{\text{CE}}(Y, g_{\text{sup}}(X_{j-1}^M))] - \mathbb{E}[l_{\text{CE}}(Y, g_{\text{sup}}(X_j^M))]. \tag{4}$$

**Summary:** The *information gain* (IG) between steps $j$ and $j-1$ reflects how much relevant information step $j$ contributes towards predicting $Y$. If task $\lambda_j$ is executed correctly, this gain is positive, as indicated by a decrease in the cross-entropy loss. Conversely, if step $j$ does not provide additional information, the loss remains unchanged. This can be interpreted as the conditional mutual information between $X_j^M$ and $Y$, conditioned on $X_{j-1}^M$. Positive information gain suggests step $j$ adds new insight about $Y$, while no gain indicates no added information. Training details for the supervisor model are in Appendix B.1.3.

**Remark on sample-wise information gain**  While conditional mutual information provides an aggregate measure of information gain for a sub-task in a dataset, it may also be desirable to obtain an analogous measure of sub-task correctness for individual CoT instances. This could be useful, for example, in detecting which step went wrong for a given prompt. Our notion of information gain can be extended to this sample-wise setting by instead considering the following difference

$$\log p(Y \mid X_j^M) - \log p(Y \mid X_{j-1}^M) \approx l_{\text{CE}}(Y, g_{\text{sup}}(X_{j-1}^M)) - l_{\text{CE}}(Y, g_{\text{sup}}(X_j^M)). \tag{5}$$

Intuitively, if step $j$ in the model's CoT is correct, the model should become more confident in the ground truth output $Y$ being the correct final answer. Therefore, the difference above should be positive. Alternatively, if step $j$ is wrong, the model's confidence regarding the true output $Y$ should not increase and the above difference should not be positive. From now on, we refer to the difference in (5) as *sample-wise information gain* at step $j$.

## 4 RELATED WORKS

**Evaluation of CoT reasoning**  Several recent works propose methodologies for evaluating CoT reasoning (Wei et al., 2024; Havrilla et al., 2024; Li et al., 2023; Joshi et al., 2023; Nguyen et al., 2024; Wang et al., 2024a; Yu et al., 2024; Xie et al., 2024). For example, Li et al. (2023) verifies individual steps in a model's CoT reasoning by generating multiple LLM responses per prompt and comparing correct responses with incorrect ones. Similarly, Wang et al. (2024b;c) use a fine-tuned LLM to decode multiple reasoning paths from each step and check the correctness of these reasoning paths. However, as we show in our experiments, approaches which simply rely on the correctness of the final output are not sound in general and can lead to false positives. Moreover, these solutions may not be plausible for problems of high difficulty where correct LLM responses might be scarce.

**Formalising CoT framework**  The formalisation of LLM reasoning remains an active area of research. Most notably González & Nori (2023) introduces a formal framework for LLMs and is a key source of inspiration behind our formalism. Additionally, Feng et al. (2023) theoretically examines the expressivity of LLMs with CoT in solving mathematical and decision-making problems, focusing on the transformer architecture's implications on accuracy. Besides this, Xu et al. (2024) provides a formal definition of hallucinations, but does not consider CoT reasoning specifically.

**Reward modelling**  One notable line of work known as outcome-based reward models (ORM) (Cobbe et al., 2021; Havrilla et al., 2024; Lightman et al., 2023) predicts the probability of reaching the correct final answer given a model's intermediate CoT steps. While ORMs do not require

demonstrations of correct intermediate steps, we show in Section 5 that this approach is not sound for detecting errors in a model's CoT reasoning. Another related method is step-wise ORM (SORM) Havrilla et al. (2024) which estimates the probability of an 'optimal' model reaching a correct answer, given the CoT reasoning of our model of interest. However, unlike our approach, SORM requires training a model which is larger and more capable than our base model.

Process-based reward modelling (PRMs) (Lightman et al., 2023; Uesato et al., 2022) is an alternative approach which directly predicts the correctness of intermediate CoT reasoning steps. Likewise, various other approaches rely on annotated CoT datasets for benchmarking (Jacovi et al., 2024; Yu et al., 2024; Amini et al., 2019; Liu et al., 2020; Xi et al., 2024; Nguyen et al., 2024; Xie et al., 2024; McLeish et al., 2024). While these benchmarks and methodologies can be valuable for improving LLM reasoning, collecting annotated data can be very costly and is not readily scalable to other tasks. Unlike these methods, our approach computes the information gain at each step, providing a richer measure of LLM performance without requiring any human-annotated CoT data.

## 5 EXPERIMENTS

In this section, we empirically demonstrate the practical utility of our framework. In addition to our proposed method dubbed information gain (denoted by IG), we consider two common baselines that can be used to detect the errors in a model's CoT reasoning and assume access to only the model's CoT generations $X_0, X_1^M, \ldots, X_T^M$ as well as the correct final answers denoted as $Y$.

**Outcome Reward Model (ORM) (Havrilla et al., 2024)**  This involves training a classifier, denoted as $f_{\text{ORM}}$, which takes as input model generations up to any step $t$ in its CoT reasoning, $X_t^M$, and predicts the probability of the model's final answer being correct, i.e.

$$f_{\text{ORM}}(X_t^M) \approx \mathbb{P}(Y^M = Y \mid X_t^M). \tag{6}$$

Here, if we observe that this probability of correctness drops significantly after step $t$, i.e. if $f_{\text{ORM}}(X_t^M) \gg f_{\text{ORM}}(X_{t+1}^M)$, this indicates that the model does not apply the task $\lambda_{t+1}$ correctly.

**Math-Shepherd (Wang et al., 2024b)**  This method quantifies the *potential* for a given reasoning process $X_t^M$ by using a 'completer' model to generate $N$ completions of each reasoning process starting from step $t$, $\{(X_t^M, X_{t+1,j}^M, \ldots, X_{T,j}^M, Y_j^M)\}_{j \leq N}$, where $Y_j^M$ denotes the final answer reached in the $j$'th completion. Then, we estimate the potential of this step based on the proportion of correct answers among the $N$ completions, denoted by $f_{\text{MS}}(X_t^M)$ as:

$$f_{\text{MS}}(X_t^M) := \sum_{j=1}^{N} \frac{\mathbb{1}(Y_j^M = Y)}{N}. \tag{7}$$

For a fair comparison we do not assume access to a 'verifier' model more capable than our base model and therefore, we use the base model as the completer model in our experiments.

### 5.1 TOY DATA EXPERIMENTS

First, we consider a toy setting where we have full control over the model behaviour on different tasks. Our prompts comprise of an integer vector $Z_0 \in \mathbb{Z}^5$ sampled randomly from a given distribution. The task $\lambda$ comprises 5-steps $\lambda = \lambda_1 \circ \ldots \circ \lambda_5$, where each sub-task $\lambda_i$ denotes an operation which transforms a given integer vector $Z_{i-1} \in \mathbb{Z}^5$ into another $Z_i \in \mathbb{Z}^5$. Finally, in this setup, the correct final answer $Y$ is the value of $Z_5$. Additional details on the data generating mechanism as well as the sub-tasks are provided in Appendix B.1.

**Generating the dataset**  To investigate partial unidentifiability for a given task $\lambda_i$ we modify the obtained dataset by introducing 'noise' at step $i$. In other words, the task $\lambda_i$ is applied incorrectly on a subset of the data, whereas all other tasks are always applied correctly. This represents a model which sometimes fails at step $i$ and we use 'LLM$_i$' to denote this model in this experiment. We repeat this procedure for all tasks $\lambda_i$ for $i \in \{1, \ldots, 5\}$ which yields 5 LLMs $\{\text{LLM}_1, \ldots, \text{LLM}_5\}$.

To also investigate the robustness of the methods, we introduce a special case in LLM$_3$. Here, task $\lambda_3$ is applied incorrectly if and only if the output after task 2 (i.e., after $\lambda_2$) lies in some set $\mathcal{S}$. This choice has been made deliberately to highlight a pitfall of the existing baselines (as we will explain

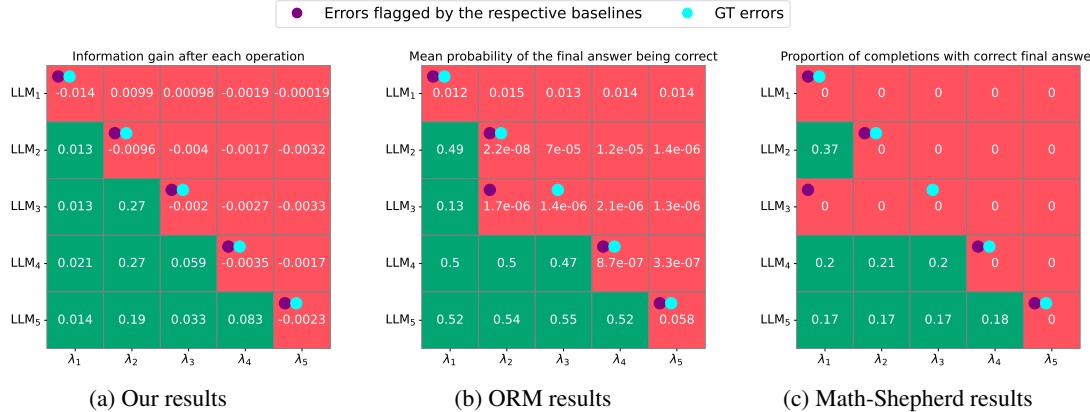

(a) Our results  (b) ORM results  (c) Math-Shepherd results

Figure 3: Heatmaps quantifying the correctness of different sub-tasks for the 5 LLMs under consideration obtained using the different baselines. Here, the red color indicates a significant drop in the plotted metrics and can be seen as an indication of an incorrectly executed sub-task.

below) and is in contrast to the rest of LLMs where any errors occur at random. In other words, the correctness of task $\lambda_3$ is dependent on the output of $\lambda_2$. For more details, see Appendix B.1.2.

### 5.1.1 RESULTS

Figure 3 shows how the different baselines quantify the correctness of the different tasks for the 5 different LLMs under consideration. This figure only considers samples where the final answer of the LLM was incorrect, i.e. $Y^M \neq Y$. For our method (IG), Figure 3a shows the information gain across the different steps for each LLM. Likewise, Figure 3b presents the results for ORM and shows how the average probability of correctness in (6) changes across the different steps, whereas, for Math-Shepherd, Figure 3c shows the proportion of correct completions starting after each step (7). Here, any significant drop in the plotted values indicate an incorrect application of a task.

**Information gain accurately quantifies step-wise correctness**  We observe that for each LLM the information gain remains positive until we encounter an incorrect reasoning step, at which point it drops to negative values. Therefore, our method can identify the incorrectly executed task for each LLM under consideration. We used a GPT-2 supervisor model to estimate information gain.

**Pitfall of the baselines**  While the ORM and Math-Shepherd manage to correctly identify the incorrect reasoning steps in most cases, these methods fail to correctly detect erroneous steps for $LLM_3$. This happens because, in our setup, $\lambda_3$ is incorrectly applied if and only if the output after task $\lambda_2$ lies in some set $\mathcal{S}$. Therefore, the classifier model can confidently predict the correctness of the final model output at $\lambda_2$ by simply checking if the output lies in $\mathcal{S}$. Here, the classifier becomes confident that the final output will be wrong right after $\lambda_2$, even though the error occurs at $\lambda_3$.

Table 1: Metrics for sample-wise classification of sub-task correctness for $LLM_3$ using the different baselines.

| METHOD | ACCURACY ↑ | TPR ↑ | FPR ↓ |
|---|---|---|---|
| IG (OURS) | **0.96** | 0.98 | **0.06** |
| ORM | 0.77 | 0.98 | 0.54 |
| MATH-SHEPHERD | 0.60 | **1.0** | 1.0 |

Similarly, when using Math-Shepherd for $LLM_3$ (with the same model being used as a completer), a completion yields an incorrect final answer if the output after $\lambda_2$ lies in $\mathcal{S}$. If this is the case, all completions yield an incorrect final output regardless of which step we begin the completions from. This makes it impossible to accurately identify the step at which $LLM_3$ goes wrong.

**Sample-wise detection**  We can also use the different baselines for sample-wise detection of erroneous steps as outlined in Section 3.3. In this setting, for each prompt, we can classify a step as incorrect if a baseline's metric falls below a threshold. Table 1 shows the results for sample-wise classification of sub-task correctness for $LLM_3$ using the different baselines (where we chose the best thresholds for each baseline using a held-out dataset). It can be seen that our method yields a significantly higher accuracy and a lower rate of false-positives than the baselines and therefore, is also considerably more reliable for sample-wise detection of errors.

## 5.2 Arithmetic operations on LLama-3-8B

Following our toy experiments, we now evaluate our framework in a more realistic setting using the Llama-3-8B model (Dubey et al., 2024). We focus on a simple arithmetic task that involves both multiplication and addition tasks. The goal is to assess the model's performance on individual operations as well as their combination.

**Experimental setup**   We sample two integers $x$ and $y$ uniformly from the range $[1, 100000)$. The prompt given to the model is structured as follows:

**Prompt:** "$x = \{x\}$, $y = \{y\}$, *Please calculate the following: 1. 3x, 2. 2y, 3. 3x + 2y* "

**Model accuracy**   We observe that the model's accuracy varies across the three steps:

Step 1 accuracy: 80%,            Step 2 accuracy: 98%,            Step 3 accuracy: 42%.

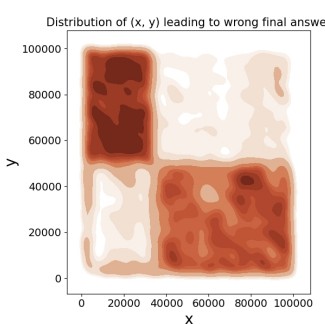

Figure 4: The distribution of $(x, y)$ for incorrect samples shows a clear trend: Llama-3-8B struggles to add large and small numbers together (top-left and bottom-right).

Notably, the majority of failures occur in the third step, which involves addition of the previously computed values. We analyzed the distribution of $(x, y)$ values where the model obtains the correct final output. Interestingly, as Figure 4 illustrates, we observed that most errors occur when exactly one of the variables $(x, y)$ is large and the other is small. This suggests that the model's correctness is highly dependent on the $(x, y)$ values in the prompt, resulting in baselines struggling to identify the erroneous step in the model's CoT reasoning (as we show below).

### 5.2.1 Results

**Our Method**   We trained the supervisor model by fine-tuning a Llama-3-8b model using Low Rank Adaptation (LoRA) (Hu et al., 2021). Table 2 shows that there is a significant drop in information gain at step 3 relative to steps 1 and 2, demonstrating that our information-theoretic method is able to correctly identify that the failure mainly occurs at step 3.

**Outcome Reward Model (ORM)**   In contrast, for ORM the mean probability of correctness included in Table 2 remains unchanged at each step. This could be explained by Figure 4 which suggests that ORM classifier can predict the correctness of the final output using only the values of $x$ and $y$ available in the prompt. Crucially, the classifier's confidence remains unchanged even as the model's intermediate reasoning steps are added to the input. Hence, ORM is unable to distinguish between the model's performance on intermediate reasoning steps.

**Math-Shepherd**   Table 2 includes the proportion of correct completions for Math-Shepherd. We observe that even though this proportion is very small at step 3, we also observe that only about 5-7% of the completions starting from steps 1 and 2 lead to a correct output, even though the error mostly occurs at step 3. This happens because the correctness of Llama-3-8B is largely determined by the initial values of $(x, y)$ in the prompt (see Figure 4). Consequently, Math-Shepherd incorrectly flags steps 1 and 2 as incorrect a significant proportion of the time which leads to a significantly higher proportion of false positives (as compared to our baseline) as we show below.

Table 2: Metrics for aggregate step-wise correctness of arithmetic operations across prompts, along with sample-wise classification of incorrect operations leading to an incorrect final answer.

|  | Step 1: $3x$ ✓ | Step 2: $2y$ ✓ | Step 3: $3x + 2y$ ✗ ‖ | Accuracy ↑ | TPR ↑ | FPR ↓ |
|---|---|---|---|---|---|---|
| IG (Ours) | 0.67 | 0.24 | 0.027 | **0.76** | 0.51 | **0.02** |
| ORM | 0.24 | 0.24 | 0.24 | 0.56 | 0.10 | 0.07 |
| Math-Shepherd | 0.068 | 0.059 | 0.00069 | 0.53 | **0.99** | 0.86 |

**Sample-wise detection**   When using these methods for sample-wise detection of incorrect steps, our approach yields the highest accuracy among the baselines considered. This superior performance is attributed to the fact that baselines like ORM and Math-Shepherd often falsely flag steps 1 and 2 as incorrect, as evidenced by their high false positive rates in Table 2.

### 5.3 Experiments on the Controlled GSM-8K Dataset

To evaluate our method on a complex dataset, we conducted experiments on GSM-8K (Cobbe et al., 2021), controlling specific factors for more interpretable results.

We begin by using GPT-4 (OpenAI et al., 2024) to generate answers for GSM-8K questions where the "multiplication" operation is always done incorrectly, while all other operations are correct. Next, we filtered the dataset to ensure that "multiplication", "subtraction", and "addition" never appeared together within the same Chain of Thought (CoT) solution. In particular, we ensured in our setting that, all incorrect final answers included both "multiplication" and "subtraction", whereas correct final answers did not involve either operation. This introduces a spurious correlation between "subtraction" and wrong answers.

In this setup, we mainly focused on evaluating ORM and our proposed method, as Math-Shepherd (with the same completer) fails trivially under these conditions. Specifically, "multiplication" is inherently unidentifiable, since any CoT containing "multiplication" negates the influence of other sub-tasks by design. Further details on the experimental setup can be found in Appendix B.3.

#### 5.3.1 Results

Table 3 demonstrates that our proposed information-theoretic approach successfully identifies the unidentifiable sub-task. Since we intentionally set the "multiplication" rules to be incorrect, we observe minimal to no information gain for this operation, as expected. However, a different pattern emerges when we examine the results of the ORM method. Both "multiplication" and "subtraction" show, on average, a very low probability of correctness. This is due to the fact that both sub-tasks are primarily associated with incorrect final answers. Consequently, relying on the standard ORM approach could lead to the misleading conclusion that "subtraction" is also incorrect.

Additionally, in our sample-wise experiment, we observe a similar trend when we use the methods to assess the sample-wise correctness of "multiplication" and "subtraction" for each prompt. Here, our proposed method not only accurately detects the unidentifiable sub-task but also highlights a significant shortcoming of ORM. Specifically, ORM falsely flags "subtraction", which is actually correct, as an incorrect sub-task due to spurious correlations.

Table 3: Comparison between our method and ORM for different sub-tasks in GSM-8K. The final three columns include results for sample-wise classification of incorrect operations for each prompt.

| | Addition ✓ | Multiplication ✗ | Division ✓ | Subtraction ✓ | | Accuracy ↑ | TPR ↑ | FPR ↓ |
|---|---|---|---|---|---|---|---|---|
| IG (Ours) | 0.99 | 0.026 | 1.05 | 1.06 | | **0.72** | 0.95 | **0.62** |
| ORM | 0.46 | 0.024 | 0.38 | 0.013 | | 0.58 | **1.0** | 1.0 |

## 6 Discussion and Limitations

In this paper, we introduce a novel information-theoretic approach for evaluating Chain-of-Thought (CoT) reasoning in large language models (LLMs) without the need for annotated intermediate steps. We present a comprehensive framework for modeling the CoT process, and the results demonstrate the effectiveness of our algorithm in identifying erroneous reasoning steps across diverse experimental settings. We consistently outperform existing baselines, including Outcome Reward Models (ORM) (Havrilla et al., 2024) and Math-Shepherd (Wang et al., 2024b) as shown in our extensive experimental section. However, it's important to note that that are some limitations to our approach.

For example, our method necessitates additional training of the supervisor model, which can be computationally demanding. Future research could investigate the use of in-context learning techniques to estimate information gain, potentially reducing the need for extra training and enhancing both the accessibility and efficiency of the approach. Secondly, sample-wise detection introduces further challenges that may lead to erroneous conclusions. A language model may occasionally arrive at the correct answer by chance, even if a particular sub-task is unidentifiable. Although this occurrence should not significantly impact the overall task-wise information gain, it could result in inaccurate outcomes for sample-wise information gain in such 'lucky' cases. Finally, while our method does not require correctness labels for every step, we still need to categorize each step according to its respective sub-task. However, this limitation is not unique to our model, as both ORM and Math-Shepherd also rely on such labels to draw sub-task-specific conclusions.

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

## A  PROOFS

*Proof of Theorem 3.3.* Suppose $\lambda$ and $\lambda'$ are two tasks with primitive decompositions

$$\lambda' = \lambda'_1 \circ \cdots \circ \lambda'_{T'}$$

and

$$\lambda = \lambda_1 \circ \cdots \circ \lambda_T, \tag{8}$$

where $\arg\min_t \{\lambda_t \notin \mathrm{Span}(\{\lambda'_1, \ldots, \lambda'_{T'}\})\} \leq k$. In other words, the primitive decompositions of $\lambda'$ and $\lambda$ diverge before step $k+1$. Then, Assumption 3.1 implies that for any $j \geq k$, we have that the answer $Y$ and $X'_j$ are d-separated by $X'_{j-1}$. Therefore,

$$Y \perp\!\!\!\perp X'_j \mid X'_{j-1}.$$

Next, we know from Assumption 3.2 that there exists some task $\hat{\lambda} \in \mathrm{Span}(\Gamma^M)$ (possibly dependent on $X_0$ and $\lambda$) such that $\Lambda^M(X_0, \lambda) \stackrel{d}{=} \Lambda(X_0, \hat{\lambda})$. Suppose that $\hat{\lambda}$ has primitive decomposition

$$\hat{\lambda} = \tilde{\lambda}_1 \circ \cdots \circ \tilde{\lambda}_{\tilde{T}},$$

then since $\hat{\lambda} \in \mathrm{Span}(\Gamma^M)$, we know that $\tilde{\lambda}_i \in \Gamma^M$ for $i \in \{1, \ldots, \tilde{T}\}$. If the primitive decomposition of $\lambda$ in (8) is such that $k = \arg\min_t \{\lambda_t \notin \mathrm{Span}(\Gamma^M)\}$, then we know that $\arg\min_t \{\lambda_t \notin \mathrm{Span}(\{\tilde{\lambda}_1, \ldots, \tilde{\lambda}_{\tilde{T}}\})\} \leq k$. Then, from the above it follows that

$$Y \perp\!\!\!\perp X^M_j \mid X^M_{j-1}.$$

Here, we used the fact that $X^M_j \stackrel{d}{=} \Lambda(X_0, \tilde{\lambda}_1 \circ \cdots \circ \tilde{\lambda}_j)$ using Assumption 3.2.

$\square$

*Proof of Proposition 3.4.*

$$\mathbb{E}[\log p(Y \mid X^M_j)] - \mathbb{E}[\log p(Y \mid X^M_{j-1})] = \mathbb{E}\left[\log \frac{p(Y \mid X^M_j)}{p(Y \mid X^M_{j-1})}\right]$$

$$= \mathbb{E}\left[\log \frac{p(Y \mid X^M_j, X^M_{j-1})}{p(Y \mid X^M_{j-1})}\right]$$

$$= \mathbb{E}\left[\log \frac{p(Y, X^M_j \mid X^M_{j-1})}{p(Y \mid X^M_{j-1})\, p(X^M_j \mid X^M_{j-1})}\right]$$

$$= \mathcal{I}(Y, X^M_j \mid X^M_{j-1})$$

Here, the second equality above arises from the fact that $X^M_j$ also captures all the information captured in $X^M_{j-1}$ (and possibly more). Therefore, conditional on $X^M_j$, the state $X^M_{j-1}$ is deterministic and hence, $Y \perp\!\!\!\perp X^M_{j-1} \mid X^M_j$.

$\square$

# B  ADDITIONAL EXPERIMENTAL DETAILS

## B.1  TOY DATA EXPERIMENTS

In this section, we describe the exact procedure used to generate the toy data for training and evaluating the models in our experiments. The dataset is constructed through five sequential operations (or tasks) applied to an initial state $z_0$, where each task $\lambda_i$ generates an intermediate state $z_i$. Both **correct** and **incorrect** examples were generated, with incorrect examples created by introducing random noise or permutations into the transformations.

The data was used to represent models $\mathrm{LLM}_1$, $\mathrm{LLM}_2$, ..., $\mathrm{LLM}_5$, each corresponding to a setting where a specific task $\lambda_i$ was partially corrupted to simulate an unidentifiable task for that model.

### B.1.1  DATA GENERATION TASKS

For each prompt, an initial 5-element vector $z_0$ was randomly sampled, and we use the notation $z_0[i]$ to denote the $i$'th component of this vector. Next, the following tasks were applied sequentially:

**Task $\lambda_1$: Pairwise Swapping**

- Correct Mapping: The first and second elements, as well as the third and fourth elements of $z_0$, are swapped:

$$z_1[0], z_1[1], z_1[2], z_1[3] = z_0[1], z_0[0], z_0[3], z_0[2]$$

- Incorrect Mapping: The entire vector is shuffled randomly.

**Task $\lambda_2$: Cumulative Summation**

- Correct Mapping: The first three elements of $z_1$ are replaced by their cumulative sum, and the fourth and fifth elements are swapped:

$$z_2 = [z_2[0], z_2[0] + z_2[1], z_2[0] + z_2[1] + z_2[2], z_1[4], z_1[3]]$$

- Incorrect Mapping: Each element of $z_1$ is perturbed by adding a random integer between 10 and 99:

$$z_2[i] = z_1[i] + U_i \quad \text{for each } i \text{ where } U_i \text{ is a randomly sampled integer between 10 and 99}$$

**Task $\lambda_3$: Reverse and Cumulative Sum**

- Correct Mapping: The first three elements of $z_2$ are reversed, and the last two elements are replaced by their cumulative sum:

$$z_3 = [z_2[2], z_2[1], z_2[0], z_2[3], z_2[3] + z_2[4]]$$

- Incorrect Mapping: As with task $\lambda_2$, each element of $z_2$ is perturbed by adding a random integer between 10 and 99.

**Task $\lambda_4$: Sorting and Elementwise Multiplication**

- Correct Mapping: The vector $z_3$ is sorted, and the first four elements are replaced by element-wise multiplications of specific pairs:

$$z_4[0] = z_3[1] \times z_3[2], \quad z_4[1] = z_3[0] \times z_3[3], \quad z_4[2] = z_3[4] \times z_3[0], \quad z_4[3] = z_3[2] \times z_3[2]$$

- Incorrect Mapping: The vector is randomly shuffled.

**Task $\lambda_5$: Difference Calculation**

- Correct Mapping: The first element is replaced by the absolute difference of the first two elements of $z_4$, and other elements are transformed as follows:

$$z_5 = [|z_4[0] - z_4[1]|, z_4[2], z_4[3], |z_4[3] - z_4[4]|, z_4[0]]$$

- Incorrect Mapping: The vector is randomly shuffled.

### B.1.2 MODELS $LLM_1, LLM_2, \ldots, LLM_5$

For each model $LLM_i$ ($i \in \{1, 2, 3, 4, 5\}$), the task $\lambda_i$ was selectively corrupted to simulate unidentifiability for that task. Specifically:

- Correct Data: The task $\lambda_i$ was applied according to its correct mapping.
- Incorrect Data: The task $\lambda_i$ was applied using its incorrect mapping (random noise, shuffling, or perturbations).

For each $LLM_i$, the tasks $\lambda_1$ to $\lambda_{i-1}$ and $\lambda_{i+1}$ to $\lambda_5$ were correctly applied, but task $\lambda_i$ was corrupted for a subset of the data. More specifically, for all LLMs except $LLM_3$, the error was introduced at step $i$ at random with probability 0.5. In contrast, for $LLM_3$, the error was introduced at step 3 if and only if the output at step 2, $z_2$ satisfies, $z_2[2] > 150$. This choice was deliberately made to highlight a pitfall of the baselines as explained in Section 5.

**String Representation of Chain-of-Thought (CoT)**   Next, we convert each sequence of vectors $z_0, z_1, \ldots, z_5$ produced by the tasks into a string-based Chain-of-Thought (CoT) representation. Each intermediate state vector $z_i$ is expressed as a comma-separated list of its elements, and the transitions between the states are delimited by "||". This format explicitly captures the step-by-step reasoning process of the model.

For example, given an initial vector $z_0 = [83, 48, 14, 98, 25]$, applying the tasks sequentially yields intermediate states $z_1, z_2, \ldots, z_5$. These states are concatenated into a single string, separated by "||" to represent the full reasoning chain:

$$
\begin{array}{c}
\texttt{83,48,14,98,25 || 48,83,98,14,25 || 48,131,229,25,14 ||} \\
\texttt{229,131,48,25,39 || 1872,3275,5725,2304,229 ||} \\
\texttt{1403,5725,2304,2075,1872}
\end{array}
$$

### B.1.3 TRAINING THE SUPERVISOR MODEL

To estimate the information gain in (3), we train a different LLM, which we refer to as the *supervisor model* $g_{\text{sup}}$. As explained in Section 3.3, this model takes as input the model's CoT reasoning up to any given intermediate step $t$, $X_t^M$, and is fine-tuned to directly predict the ground truth final output $Y$. To this end, we use a special token to separate the model's CoT reasoning and the final output when fine-tuning $g_{\text{sup}}$. At inference time, this special token when appended to the model input serves as an indication for the model to directly predict the final output. In this way $g_{\text{sup}}(X_t^M)$ approximates the conditional distribution $p(Y \mid X_t^M)$.

More specifically, in the toy setup discussed above, consider the following sample for model's CoT:

$$
\begin{array}{c}
\texttt{83,48,14,98,25 || 48,83,98,14,25 || 48,131,229,25,14 ||} \\
\texttt{229,131,48,25,39 || 1872,3275,5725,2304,229 ||} \\
\texttt{1403,5725,2304,2075,1872}
\end{array}
$$

For this example, the ground truth final output $y$ is $y = $ "$\texttt{1403,5725,2304,2075,1872}$" (i.e., the model reached the correct final output in the example above).

For the sample given above, we have that

$x_0^M = x_0 = $ "$\texttt{83,48,14,98,25}$"

$x_1^M = $ "$\texttt{83,48,14,98,25 || 48,83,98,14,25}$"

$\qquad \vdots$

$x_5^M = $ "$\texttt{83,48,14,98,25 || 48,83,98,14,25 || 48,131,229,25,14 ||}$
$\qquad \texttt{229,131,48,25,39 || 1872,3275,5725,2304,229 ||}$
$\qquad \texttt{1403,5725,2304,2075,1872}$"

Next, to construct the data for fine-tuning the supervisor model, we used the special token "$\texttt{\#|>}$" to separate the model's CoT steps $x_i^M$ from the ground truth output $y$. This results in the following 6 training datapoints for the supervisor model:

1. "$\texttt{83,48,14,98,25 \#|> 1403,5725,2304,2075,1872}$"

2. "$\texttt{83,48,14,98,25 || 48,83,98,14,25 \#|> 1403,5725,2304,2075,1872}$"

$\qquad \vdots$

5. "$\texttt{83,48,14,98,25 || 48,83,98,14,25 || 48,131,229,25,14}$
$\qquad \texttt{|| 229,131,48,25,39 || 1872,3275,5725,2304,229 ||}$
$\qquad \texttt{1403,5725,2304,2075,1872 \#|> 1403,5725,2304,2075,1872}$"

The above procedure allows us to obtain fine-tuning data for supervisor models separately for each of the 5 different LLMs, $\{\text{LLM}_1, \text{LLM}_2, \ldots, \text{LLM}_5\}$. Next, we train a separate GPT-2 model for each of the 5 different base LLMs.

### B.1.4 ESTIMATING THE INFORMATION GAIN

Having trained the supervisor model on the data generated above, we evaluate the information gain on a held-out dataset split. Given a datapoint $(x_i^M, y)$ in the evaluation split, we can estimate the sample-wise information gain at step $i$ as follows:

- Suppose that the model generation at step $i - 1$, $x_{i-1}^M$ is tokenised as $(t_1, \ldots, t_{n_{i-1}})$ and similarly that $x_i^M$ is tokenised as $(t_1, \ldots, t_{n_i})$. Likewise, suppose that the true output $y$ is tokenised as $(t_1^*, \ldots, t_k^*)$ and we use $< s >$ to denote the separator token (i.e. # | > above).

- Then, to estimate the sample-wise for this datapoint, we estimate the difference:

$$\frac{1}{k} \sum_{j=1}^k \log p(t_j^* \mid (t_1, \ldots, t_{n_i}, < s >, t_1^*, \ldots, t_{j-1}^*))$$

$$- \frac{1}{k} \sum_{j=1}^k \log p(t_j^* \mid (t_1, \ldots, t_{n_{i-1}}, < s >, t_1^*, \ldots, t_{j-1}^*)).$$

  Here, the supervisor model is trained to estimate the above conditional and therefore we use it to estimate the difference above.

Finally, to estimate the aggregate information gain (instead of the sample-wise information gain), we simply compute the average sample-wise gain over the evaluation data split.

### B.1.5 ADDITIONAL RESULTS

In Figures 5 - 7, we present the sample-wise trajectories for 15 randomly chosen prompts leading to incorrect final answers, for the different baselines and LLMs under consideration. Here, any significant drop in the plotted value at a given step could be seen as an indication of an incorrectly executed sub-task. Recall that in our setup, in $\text{LLM}_i$, the CoT step $i$ is executed incorrectly with some probability whereas all other steps are always executed correctly.

Firstly, Figure 5 presents sample-wise information gain for our method for the five different LLMs. Here, we see that the sample-wise information remains high up until the incorrect step, at which point the information gain sharply decreases. This suggests that sample-wise information gain is sensitive to the specific point where the Chain of Thought goes wrong, making it effective at locating reasoning errors.

For the ORM and Math-Shepherd baselines in Figures 6 and 7, we observe that for all LLMs except $\text{LLM}_3$, the plotted metrics drop at the incorrect step. However, for $\text{LLM}_3$, we observe that ORM's probability of correctness drops at step 2 even though the error occurs at step 3. This occurs because, in our setup, the correctness of step 3 is determined directly from the output of step 2. Specifically, recall that in $\text{LLM}_3$, step 3 is executed incorrectly if and only if the output of step 2, $z_2$, has its second component greater than 150, i.e. $z_2[2] > 150$. Therefore, ORM becomes confident after the second step if a CoT is going to lead towards the correct final answer or not.

Similarly, for Math-Shepherd in Figure 7, we observe that the proportion of correct completions remains 0 for $\text{LLM}_3$. This is because for all trajectories plotted, the output of step 2, $z_2$, has its second component greater than 150 and therefore the final answer is incorrect regardless of which step we begin the completions from.

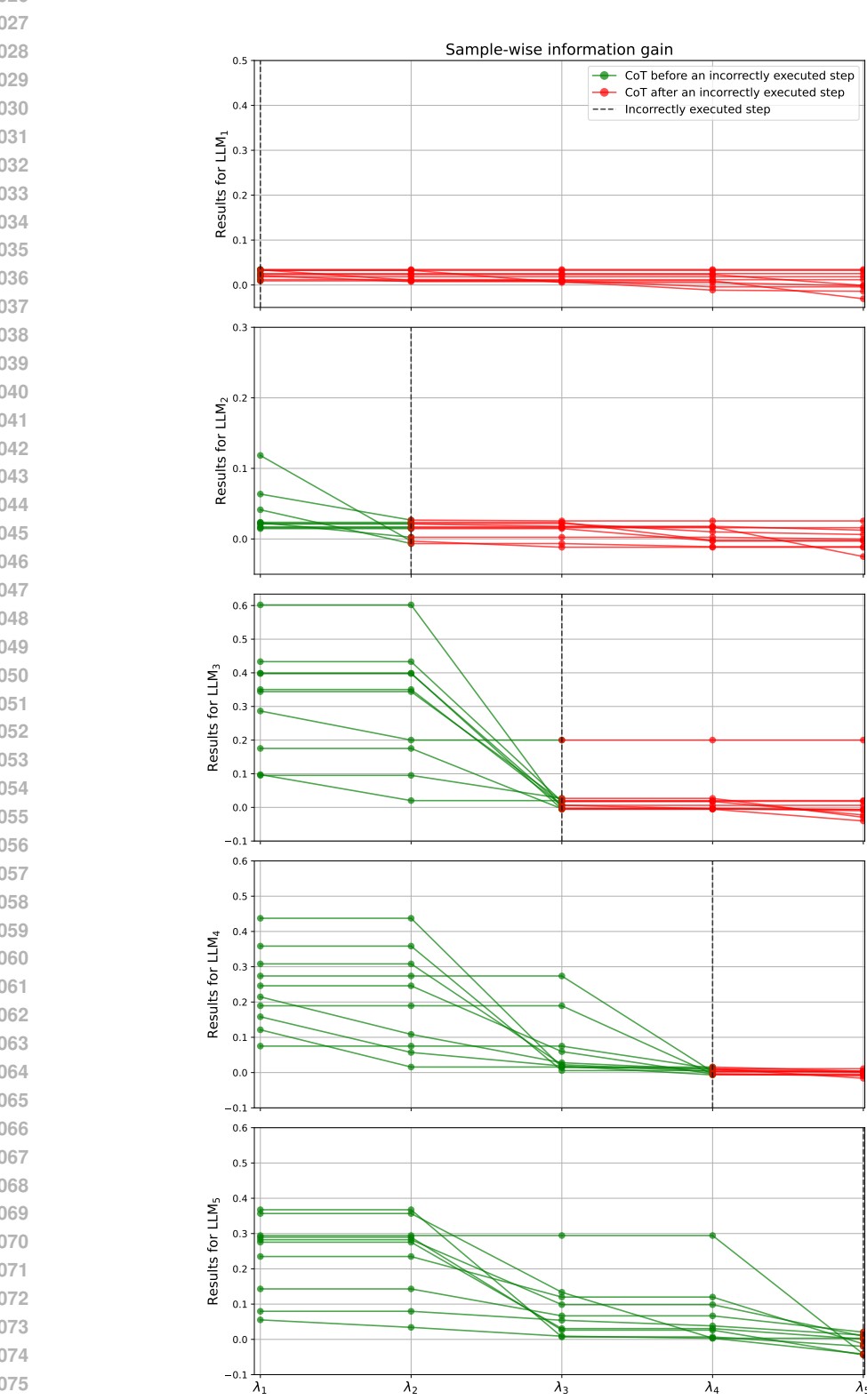

Figure 5: Toy data results: Sample-wise information gain trajectories for 15 randomly chosen prompts with wrong final answers.

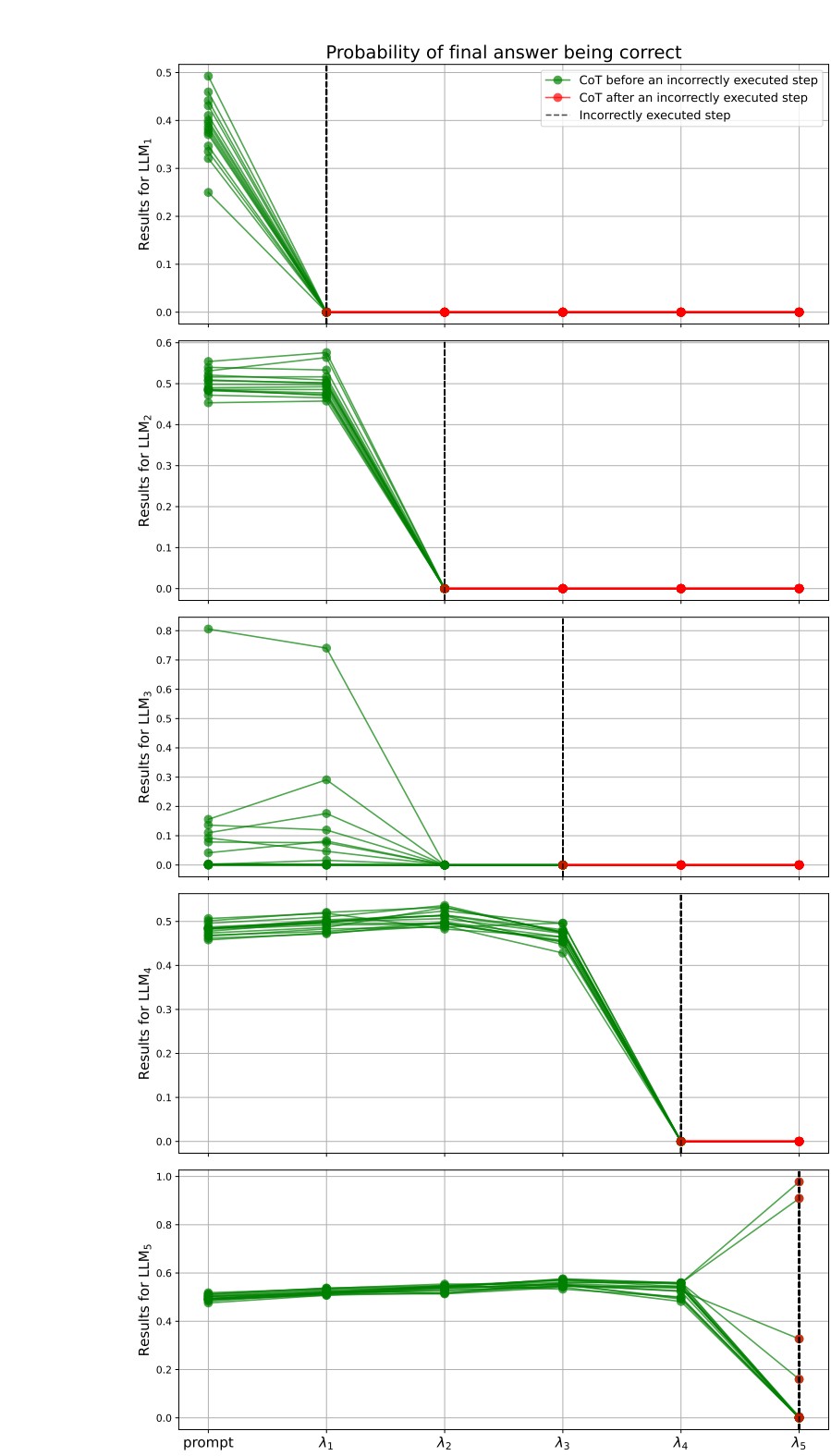

Figure 6: Toy data results: ORM's probability of correctness after each step for 15 randomly chosen prompts with wrong final answers

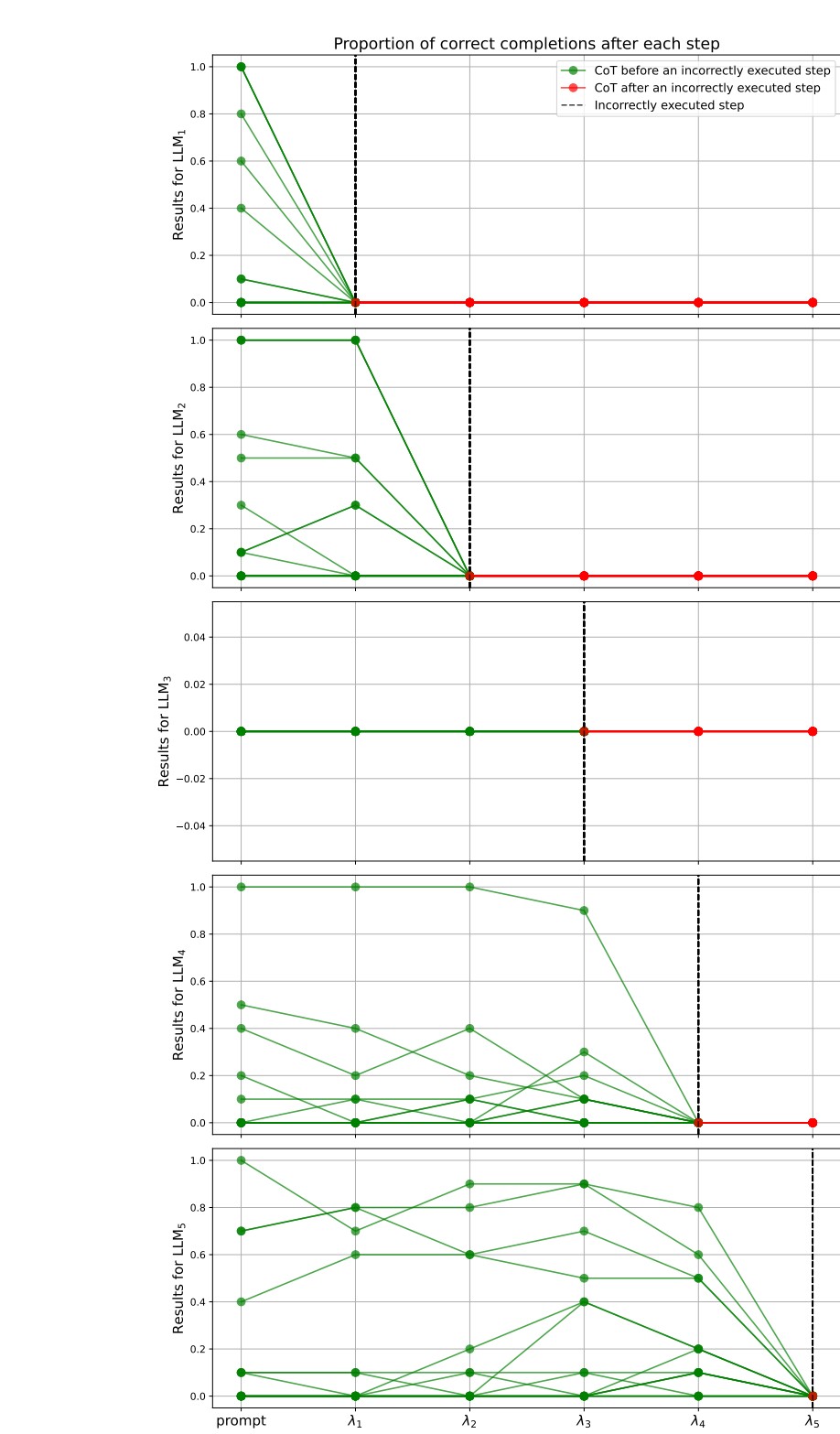

Figure 7: Toy data results: Math-Shepherd's proportion of correct completions from each step for 15 randomly chosen prompts with wrong final answers

## B.2 ARITHMETIC OPERATIONS ON LLAMA 3 8B

For this experiment, the prompts used to collect the data follow a specific structure. Each prompt contains two real examples followed by a query with newly sampled values for $x$ and $y$. The format of the prompt is as follows:

```
x = 23, y = 51. Please calculate the following:
1. 3x
2. 2y
3. 3x + 2y
Answer:
1. 3x = 69
2. 2y = 102
3. 3x + 2y = 171

x = 35, y = 60. Please calculate the following:
1. 3x
2. 2y
3. 3x + 2y
Answer:
1. 3x = 105
2. 2y = 120
3. 3x + 2y = 225

x = {x}, y = {y}. Please calculate the following:
1. 3x
2. 2y
3. 3x + 2y
Answer:
```

In the third section, the values of $x$ and $y$ are randomly sampled from a uniform distribution over the range $[1, 100000)$.

### B.2.1 TRAINING DATA FOR THE SUPERVISOR MODEL

The *supervisor model* plays a crucial role in evaluating the intermediate steps in the Chain-of-Thought (CoT) reasoning. The model is designed to approximate the probability of arriving at the correct final result after any given step in the CoT process. To train this model, we fine-tune it using a dataset composed of generated CoT steps concatenated with the correct final result.

**Model Generation Example:** Consider the following example of a model-generated response:

```
x = 51290.0, y = 90718.0. Please calculate the following:
1. 3x
2. 2y
3. 3x + 2y
Answer:
1. 3x = 153770.0
2. 2y = 181436.0
3. 3x + 2y = 335206.0
```

**Fine-Tuning Data Construction:** The generated outputs are used to construct training examples, where each intermediate step is concatenated with the final correct answer using the separator token `#|>`. For instance, from the example above, the following four training data points are created:

1. `"x = 51290.0, y = 90718.0. Please calculate the following: 1. 3x 2. 2y 3. 3x + 2y Answer: #|> 3x + 2y = 335306.0"`
2. `"x = 51290.0, y = 90718.0. Please calculate the following: 1. 3x 2. 2y 3. 3x + 2y Answer: || 1. 3x = 153770.0 #|> 3x + 2y = 335306.0"`

```
3. "x = 51290.0, y = 90718.0.  Please calculate the following:
   1.  3x 2.  2y 3.  3x + 2y Answer:  || 1.  3x = 153770.0 ||
   2.  2y = 181436.0 #|> 3x + 2y = 335306.0"

4. "x = 51290.0, y = 90718.0.  Please calculate the following:
   1.  3x 2.  2y 3.  3x + 2y Answer:  || 1.  3x = 153770.0 ||
   2.  2y = 181436.0 || 3.  3x + 2y = 335206.0 #|> 3x + 2y =
   335306.0"
```

Each step concatenates the current state of reasoning with the correct final answer. This process enables the supervisor model to learn the relationship between intermediate steps and the correct final outcome.

Finally, using the dataset generated above, we fine-tune a Llama-3-8b model using Low Rank Adaptation (LoRA) (Hu et al., 2021) as the supervisor model. Finally, the information gain is computed using the trained model as described in Section B.1.4.

### B.2.2 MATH SHEPHERD RESULTS

The Math-Shepherd approach (Wang et al., 2024b) evaluates how well the model generates intermediate results and completes the reasoning process step-by-step. For a given model generation, we iteratively cut off the chain of reasoning after each step and obtain multiple completions using a completer model (in this case, also the Llama-3-8B model).

Consider the following model generation:

```
x = 51290.0, y = 90718.0. Please calculate the following:
1. 3x
2. 2y
3. 3x + 2y
Answer: 1. 3x = 153770.0, 2. 2y = 181436.0, 3. 3x + 2y = 335206.0
```

In this example, the model completes the full sequence of steps for $x = 51290.0$ and $y = 90718.0$. To assess the robustness of the Chain-of-Thought (CoT) process, we perform the following procedure for the Math Shepherd results:

1. Step-wise Completion: We cut off the generation after each step in the reasoning process. For instance, after computing $3x = 153770.0$, we stop the generation there and generate 10 completions using the Llama-3-8b model.

2. Multiple Completions: At each cut-off point, the Llama-3-8b model is tasked with completing the remaining steps of the chain of reasoning. For each step, 10 independent completions are generated.

3. Proportion of Correct Completions: For each cut-off point, we compute the proportion of correct completions. This proportion gives insight into how likely the model is to complete the remaining steps of reasoning correctly, starting from the intermediate point. For example, after cutting off the reasoning at $3x = 153770.0$, we evaluate how many of the 10 completions successfully compute $3x + 2y = 335306.0$.

In this way, Math-Shepherd quantifies the model's ability to continue reasoning correctly at each intermediate stage.

### B.2.3 ADDITIONAL RESULTS

Figures 8 - 10 present the sample-wise trajectories for 15 randomly chosen prompts leading to incorrect final answers for the different baselines. Here, once again, any significant drop in the plotted value at a given step could be seen as an indication of an incorrectly executed sub-task. Recall that in this setup majority of the errors occur at the final step which involves the addition of $3x + 2y$.

Figure 8 shows the sample-wise information gain for our method after each step. We see that for most of the plotted trajectories, the sample-wise information gain remains high until the final step,

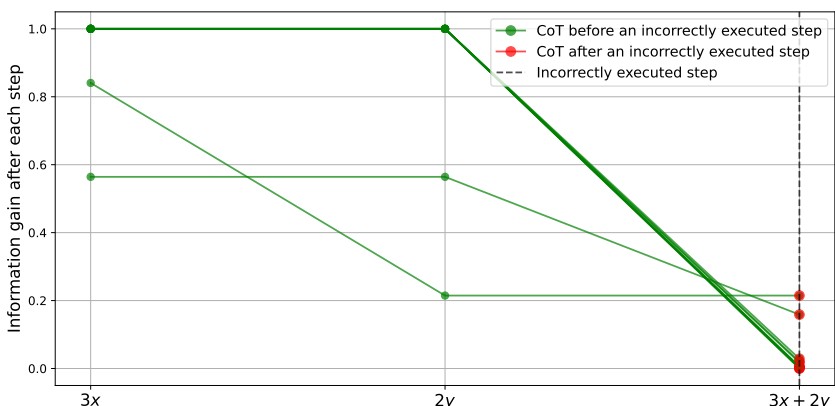

Figure 8: Arithmetic operations on Llama-3-8b: Sample-wise information gain trajectories for 15 randomly chosen prompts with wrong final answers

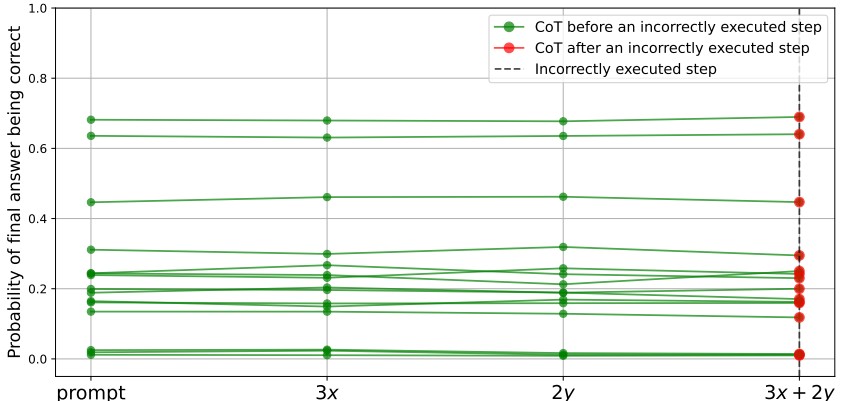

Figure 9: Arithmetic operations on Llama-3-8b: ORM's probability of correctness after each step for 15 randomly chosen prompts with wrong final answers

at which point it drops to values close to or below 0. This shows that our method correctly identifies that the failure predominantly occurs at step 3.

In contrast, Figure 9 shows that the mean probability of correctness for the ORM remains unchanged at each step. This could be explained by Figure 4 in the main text, which suggests that the ORM classifier can predict the correctness of the final output using only the values of $x$ and $y$ available in the prompt. Crucially, the classifier's confidence remains unchanged even as the model's intermediate reasoning steps are added to the input. This means that ORM is unable to distinguish between the model's performance on intermediate reasoning steps.

For Math-Shepherd results shown in Figure 10, most of the trajectories plotted remain constant at 0. In other words, when using Llama-3-8B as the completer model, we observe that for most of the prompts, no completion leads to the correct answer, regardless of which step we begin the completions from. This is likely because, for most of the examples considered in this plot, the $(x, y)$ combination in the prompt has exactly one small value and the other is large (as shown in Figure 4). This also highlights why Math-Shepherd has a high false positive rate.

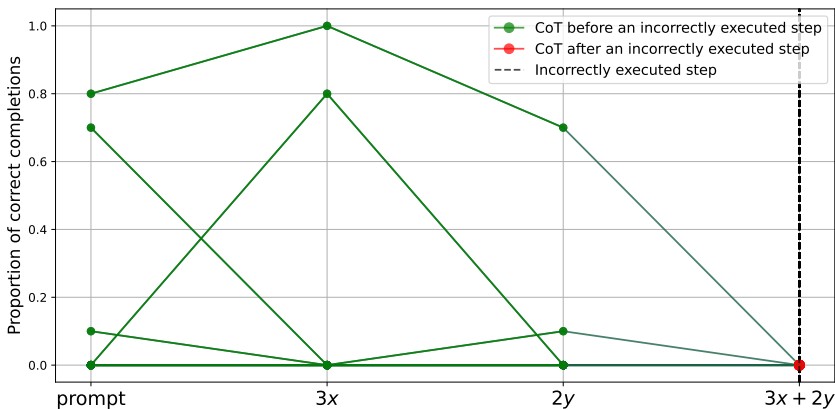

Figure 10: Arithmetic operations on Llama-3-8b: Math-Shepherd's proportion of correct completions from each step for 15 randomly chosen prompts with wrong final answers

### B.3 Controlled GSM-8K Experiments

In order to understand if our proposed method also works on more textual data, we set out to perform an experiment on the popular GSM-8K dataset which has more elaborate prompts compared to the previous experiments. To this end, we follow these steps:

- We first construct the dataset by using the GPT-4 API on the question. This will give us the basis for correct CoTs.

- Next we also again use GPT-4 to label each of the intermediate steps as either using "Addition", "Subtraction", "Division" or "Multiplication".

- With this data in hand, we can now construct our unidentifiable operations. In particular, we again use GPT-4 to rewrite all the CoTs which contain a multiplication into CoTs, where the multiplication is performed wrong and subsequently also the final answer is wrong.

- Finally, we filter the final dataset, where we make sure that for every failed CoT, we have both "multiplication" and "subtraction" in the CoT, and for every correct CoT, we do not have these operations. This leads us to a setting where standard methods such as ORM would likely fail as there is no way to distinguish between multiplication and subtraction for the classifier.

#### B.3.1 Examples

Here below are some examples from the dataset, where we start with the prompt, followed by the CoT generations (with a wrong multiplication).

**Prompt 1:**

```
Julie is reading a 120-page book.  Yesterday, she was able to read
12 pages and today, she read twice as many pages as yesterday.  If
she wants to read half of the remaining pages tomorrow, how many
pages should she read?
```

**Model response 1:**

```
Calculate today's pages read by doubling yesterday's pages:
12 * 2 = 46.  Calculate the total pages read so far by adding
yesterday's and today's pages:  12 + 46 = 58.  Determine the
remaining pages by subtracting the pages read so far from the
total:  120 - 58 = 62.  Calculate the number of pages Julie should
read tomorrow by taking half of the remaining pages:  62 / 2 = 31.
```

**Prompt 2:**

In a conference room, 40 chairs with a capacity of 2 people each
were arranged in rows in preparation for the board meeting of
a company, whose number of members was the same as the chairs'
capacity.  If 2/5 of the chairs were not occupied, and the rest
each had two people, calculate the number of board members who did
attend the meeting.

**Model response 2:**

Calculate the total capacity of the conference room:  40 chairs *
2 people per chair = 134 people.  Calculate the number of chairs
not occupied:  40 chairs * 2/5 = 16 chairs.  Calculate the number
of chairs occupied:  Total number of chairs – Chairs not occupied
= 40 chairs – 16 chairs = 24 chairs.  Calculate the number of
people that attended (each occupied chair has 2 people):  24
chairs * 2 people per chair = 63 people.

**Prompt 3:**

Dan owns an ice cream shop and every sixth customer gets a free
ice cream cone.  Cones cost $2 each.  If he sold $100 worth of
cones, how many free ones did he give away?

**Model response 3:**

Calculate the number of cones sold that were not free by dividing
the total sales by the cost per cone:  100 / 2 = 50.  Calculate
the total number of cones including free ones by assuming every
sixth cone is free, so for every 5 sold, there is 1 free.  Total
cones = (5 sold + 1 free) * (cones sold / 5) = 6 * (50 / 5) = 6
* 10 = 72.  Calculate the number of free cones given away:  Total
cones – Cones sold = 72 – 50 = 22.

### B.3.2 TRIANING DATA FOR THE SUPERVISOR MODEL

For our supervisor model, we simply use a GPT-2 model that we SFT until convergence and use
early stopping based on a held out validation dataset. The training data for this model is composed
of generated CoT steps concatenated with the correct final output (as in other experiments).

For example, consider prompt 3 and its response above. For this prompt, the correct final response
is 10. Using this prompt, we generate 4 training datapoints for the supervisor model by truncating
the response at each step and concatenating the correct final answer using the separator token '#|>'.

1. Dan owns an ice cream shop and every sixth customer gets a
   free ice cream cone.  Cones cost $2 each.  If he sold $100
   worth of cones, how many free ones did he give away?  #|> 10

2. Dan owns an ice cream shop and every sixth customer gets a
   free ice cream cone.  Cones cost $2 each.  If he sold $100
   worth of cones, how many free ones did he give away?  ||
   Calculate the number of cones sold that were not free by
   dividing the total sales by the cost per cone:  100 / 2 = 50
   #|> 10

3. Dan owns an ice cream shop and every sixth customer gets a
   free ice cream cone.  Cones cost $2 each.  If he sold $100
   worth of cones, how many free ones did he give away?  ||
   Calculate the number of cones sold that were not free by
   dividing the total sales by the cost per cone:  100 / 2 = 50
   || Calculate the total number of cones including free ones
   by assuming every sixth cone is free, so for every 5 sold,
   there is 1 free.  Total cones = (5 sold + 1 free) * (cones
   sold / 5) = 6 * (50 / 5) = 6 * 10 = 72 #|> 10

4. ```
Dan owns an ice cream shop and every sixth customer gets a
   free ice cream cone.  Cones cost $2 each.  If he sold $100
   worth of cones, how many free ones did he give away?  ||
   Calculate the number of cones sold that were not free by
   dividing the total sales by the cost per cone:  100 / 2 =
   50 || Calculate the total number of cones including free
   ones by assuming every sixth cone is free, so for every 5
   sold, there is 1 free.  Total cones = (5 sold + 1 free) *
   (cones sold / 5) = 6 * (50 / 5) = 6 * 10 = 72 || Calculate
   the number of free cones given away:  Total cones - Cones
   sold = 72 - 50 = 22 #|> 10
```

### B.3.3  ESTIMATING THE INFORMATION GAIN

Our procedure for estimating the information gain is very similar to that described in Section B.1.4. However, in this setup, there is no fixed ordering of tasks for all prompts. For instance, in some prompts, the first step might be addition while in others it might be multiplication. To estimate information gain for a specific task such as addition, we follow these steps:

- We first consider all prompts which contain addition as a sub-task.
- Next, for these prompts we estimate the $\mathbb{E}[\log p(Y \mid X_{T_+}^M)]$ term, where $T_+$ denotes the step at which addition is executed.
- Similarly, we estimate the $\mathbb{E}[\log p(Y \mid X_{T_+-1}^M)]$ term, where $T_+ - 1$ denotes the step immediately preceding addition.
- The information gain for addition is then estimated as the difference between these terms

$$\mathbb{E}[\log p(Y \mid X_{T_+}^M)] - \mathbb{E}[\log p(Y \mid X_{T_+-1}^M)].$$

