# OpenReview forum: "Understanding Chain-of-Thought in LLMs Through Information Theory"
_ICLR.cc/2025/Conference — Submitted to ICLR 2025_

### Official Review · Reviewer_mocU · 2024-10-27

**Soundness:** 3
**Presentation:** 3
**Contribution:** 2
**Rating:** 8
**Confidence:** 2

**Summary:**

The paper proposes a novel, information-theoretic framework to analyze Chain-of-Thought (CoT) reasoning in large language models (LLMs). The authors focus on quantifying the "information gain" at each reasoning step, offering a method to detect errors in intermediate steps without requiring human-annotated CoT data. The framework uses concepts like task unidentifiability and conditional mutual information to identify at which intermediate step an LLM fails to execute a particular sub-task during a CoT process. The paper introduces a practical algorithm based on this framework to offer a more granular evaluation of LLM performance on reasoning tasks. Extensive experiments are conducted on toy datasets and the GSM-8K benchmark, showing that this method outperforms existing baselines such as Outcome Reward Models (ORM) and Math-Shepherd in detecting errors and providing insights into model performance.

**Strengths:**

1. The paper presents a novel approach to evaluating CoT reasoning by using information theory, specifically through the concept of information gain at each reasoning step. Compared to traditional methods like ORM, which focus on final outcome, the focus on intermediate reasoning steps allows for a more granular analysis of where a model might be going wrong which is important for LLM reasoning.

2. The authors provide clear mathematical foundations and definitions for the formulation of their framework. The methodology is well-structured, and the theoretical assumptions are systematically translated into a practical algorithm. The paper’s thorough empirical validation using toy data and GSM-8K provides strong evidence of the framework’s effectiveness in identifying sub-task failures.

3.  By offering a means to detect specific reasoning failures without annotated data, this approach could reduce the need for expensive human supervision and help improve the interpretability and reliability of LLMs in complex reasoning tasks.

**Weaknesses:**

1. In section 3.3, while the information-theoretic approach is innovative, the need to train a separate supervisor model to estimate information gain might introduce complexity. The paper states that the framework doesn't require annotated data, but the supervisor also requires finetune to predict the ground truth final output.
2. In section 2.2, The framework relies on the decomposition of tasks into primitive sub-tasks, and the assumption that each primitive task contributes incrementally to the information gain. For complex reasoning tasks with intertwined dependencies between sub-tasks, the information gain for each step will be more complex, and this assumption might not always hold.
3. There are some errors in the paper, for example, in appendix B.1.1, the annotation for task $\lambda_2$, if I understand correctly, should be $z_1[0], z_1[0]+z_1[1], z_1[0]+z_1[1]+z_1[2]$.

**Questions:**

1. For math problems, the primitive tasks are intuitive to define. Have you tested your framework with other tasks, e.g. Blocksworld, or commonsense QA, where the definition of primitive tasks is less straightforward? Will different primitive task affect the conclusion?
2. In section 5.1.1, you mentioned the evaluation only considers samples where the final answer of the LLM was incorrect. Could you explain why you apply such a setting since it's also a common failure for CoT to get a correct answer but with incorrect intermediate steps?
3.  Also in section 5.1.1, when discussing about pitfall of the baselines, for ORM, you stated that the classifier becomes confident that the final output will be wrong right after $\lambda_2$, even though the error occurs at $\lambda_3$. But this is what we expect from the error detection, right? I understand you want to show the classifier will overfit on learning the false correlation between $z_2[2] > 150$ and the final error as in section 5.2. But this statement/setting doesn't seem correct to me.

---

> ### Author Response · Authors · 2024-11-18
> **Rebuttal**
>
> First of all, we would like to thank the reviewer for their thorough feedback on our paper. We would also like to thank the reviewer for praising our proposed framework as a "novel approach to evaluating CoT reasoning by using information theory", being "well-structured, and the theoretical assumptions are systematically translated into a practical algorithm." Below, we will address any comments that were raised by the reviewer.
>
> > In section 3.3, while the information-theoretic approach is innovative, the need to train a separate supervisor model to estimate information gain might introduce complexity. The paper states that the framework doesn't require annotated data, but the supervisor also requires finetune to predict the ground truth final output.
>
> We would like to clarify that the data required in our setting comprises only the prompts as well as corresponding correct final answers and hence do not require any annotations of intermediate steps (as with standard Process reward models). In our case, the supervisor model takes the CoT steps and is only finetuned to predict the final answer from each step. Note that this means that we do not require any human annotations for the correctness of each intermediate step, rather only a decomposition which is natural in CoT.  We will clarify this in the final version of our manuscript.
>
> > In section 2.2, The framework relies on the decomposition of tasks into primitive sub-tasks and the assumption that each primitive task contributes incrementally to the information gain. For complex reasoning tasks with intertwined dependencies between sub-tasks, the information gain for each step will be more complex, and this assumption might not always hold.
>
> We thank the reviewer for raising this interesting point. Firstly, we would like to clarify that it is not a theoretical assumption in our framework that each intermediate CoT step contributes incrementally to the information gain. In fact, this follows directly from Proposition 3.4, which establishes this property theoretically without requiring any additional assumptions.
>
> However, we agree with the reviewer that in practice in complex settings where the CoT may involve a large number of intertwined intermediate steps, estimating the information gain accurately can become more challenging due to the increased complexity of the dataset and task structure. In such scenarios, this difficulty could be mitigated by obtaining additional CoT data for training the supervisor model, thereby providing better coverage of the complex dependencies.
>
> Additionally, using larger and more expressive supervisor models may further enhance the framework's ability to process such intricate data effectively. Note that in our experiments, we found that the relatively small GPT-2 models worked well as supervisor models, and we only required relatively small amounts of training data for fine-tuning these models (e.g., roughly 2000 samples were used to fine-tune our supervisor model for GSM-8K experiments).
> We appreciate the reviewer’s suggestion, and we will include a discussion of this point in the final version of our paper to address these considerations more explicitly.
>
> > There are some errors in the paper, for example, in appendix B.1.1, the annotation for task λ2, if I understand correctly, should be z1[0],z1[0]+z1[1],z1[0]+z1[1]+z1[2].
>
> Thanks! We thank the reviewer for having gone through our paper so thoroughly and will correct this typo in the final version of this paper.

---

> > ### Comment · Reviewer_mocU · 2024-11-21
> > **Answer to the rebuttal**
> >
> > Thanks for the clarification. The explanation for decomposition in more complex settings makes things clear. And it's good that the only data you use for training is the prompts. Using the reasoning chain from CoT as steps makes sense in this setting. I'd raise my rating accordingly.

---

> ### Author Response · Authors · 2024-11-18
> **Rebuttal (Continued)**
>
> > For math problems, the primitive tasks are intuitive to define. Have you tested your framework with other tasks, e.g. Blocksworld, or commonsense QA, where the definition of primitive tasks is less straightforward? Will different primitive task affect the conclusion?
>
> This is indeed a very interesting suggestion, and we did consider this direction. However, given the scope of this paper, we focused primarily on the foundational aspects of our framework, particularly in the context of mathematical reasoning, as these settings are easier to control and analyze systematically.
>
> For Blocksworld, the main task is "planning," which itself can be decomposed into sub-tasks comprising specific sequences or tactics of primitive actions (e.g., "stack," "unstack," "move"). In this setting, our information-gain methodology could be applied to evaluate LLM's performance in planning these sub-tasks. By examining the information gain for each sub-task, we could assess where the reasoning process succeeds or fails.
>
> Similarly, in Commonsense QA, the CoT steps could be categorized into distinct reasoning types, such as causal reasoning (identifying cause-and-effect relationships), temporal reasoning (understanding the sequence and timing of events), or spatial reasoning (comprehending physical arrangements and relationships between objects). These categories align naturally with our framework, enabling an evaluation of the LLM's reasoning steps within each category.
>
> While these tasks lend themselves well to our framework, we acknowledge that they may introduce additional nuances or challenges, particularly concerning the assumptions underlying our methodology. Exploring such tasks in detail is indeed an exciting direction which we leave for future work.
>
> > [...] a common failure for CoT to get a correct answer but with incorrect intermediate steps?
>
> We thank the reviewer for pointing out this special case that we mentioned in the limitations of our paper. However, we would like to clarify a few things regarding the setup where the LLMs get the correct answer with incorrect intermediate steps.
>
> Recall that our method only estimates the relevant information for the correct final answer added by each successive step. Therefore, in cases where an intermediate step is incorrect, it should add no useful information regarding the correct final answer and therefore the information gain at that step should still be 0 (regardless of whether the final output is correct or not).
> In contrast, in cases where the model indeed consistently commits errors at intermediate steps which then consistently lead to the correct answer, the baselines such as ORM and Math Shepherd would fail to identify such errors as these methods mainly rely on the correctness of the final answer.
>
> Nevertheless, your question did spark some interest in us and hence we went back to our datasets to check how often this indeed happens and report that in our arithmetic experiments in Section 5.2 only 1.2% of the samples indeed exhibit this behaviour where an intermediate step is wrong, but the final answer is correct. For the synthetic toy experiment, this never happens as we have control over the generation to some degree.
> Hence, we would argue that for our settings, these are relatively rare and would not affect the outcomes of our framework. We thank the reviewer again for their insightful comment to improve our paper and will definitely add a discussion on this in the final version of our paper.
>
>
> > Also in section 5.1.1, when discussing about pitfall of the baselines, for ORM, you stated that the classifier becomes confident that the final output will be wrong right after λ2, even though the error occurs at λ3. But this is what we expect from the error detection, right? I understand you want to show the classifier will overfit on learning the false correlation between z2[2]>150 and the final error as in section 5.2. But this statement/setting doesn't seem correct to me.
>
> In Section 5.1.1 we aimed to understand if the respective methods were able to correctly identify if a rule $\lambda_i$ had been incorrectly applied. We agree that this behaviour is expected as ORM will learn the correlational relationship between step 2 and the final answer, i.e., the ORM model will become confident that the output is wrong as soon as the sub-task $\lambda_2$ is applied. Therefore, if we were using the ORM model's confidence to debug which sub-task is applied incorrectly, we would be misled into concluding that step $\lambda_2$ is applied incorrectly, even though the error arises at step $\lambda_3$. We will ensure that this is properly discussed in our final version of the paper.
>
> Lastly, we would like to thank the reviewer again for their insightful comments and thorough reading to improve our paper. We hope that the above has addressed any outstanding questions and that the reviewer would consider raising their score if all the questions have been appropriately answered.

---

### Official Review · Reviewer_h9ff · 2024-11-01

**Soundness:** 3
**Presentation:** 3
**Contribution:** 3
**Rating:** 5
**Confidence:** 4

**Summary:**

This paper proposes a new method to understand the effectiveness of chain-of-thoughts. Specifically, The paper proposes to analyze CoT from the information theory perspective. By training a model to predict the information gain, the proposed method could analyze the quality of generated thinking steps and thus select the better answer. The theoretical formulation is intuitive, but more empirical challenges are overlooked. I think more experiments on the general-purpose datasets are needed to justify the value of the proposed framework. Please refer to my questions for more details.

**Strengths:**

1. The authors give a thorough and detailed rhetorical formulation of the proposed solution.
2. The paper is clearly written and easy to follow.

**Weaknesses:**

1. The gap between the proposed framework and the claimed application scenarios remains big, which affects the potential impact of this work.
2. Some of the terms are not clearly defined, especially in the LLM scenarios. (e.g., steps)
Please refer to the questions section for more detailed comments.

**Questions:**

1. Section 2.1. Do we need to, and how to define the set of lambda in a realistic task?
2. How do you define a step in a generation task?
3. It seems like the key is modeling the information gain. However, a separate model approximates the conditional distribution in the proposed framework, which is then used to model the information gain. As we all know, the training of a separate model is determined by the training data and sampled trajectories. How do you guarantee the learned model can model the conditional distribution well? In other words, what is the requirement for training such a model? Do we need to guarantee that the training data of the separate model follows a similar distribution of the final task?

---

> ### Author Response · Authors · 2024-11-18
> **Rebuttal**
>
> We appreciate the reviewer’s recognition of the clarity of our presentation and the thoroughness of our theoretical framework, as well as their constructive comments. Below, we address each of the points raised in the review.
>
> > The gap between the proposed framework and the claimed application scenarios remains big, which affects the potential impact of this work.
>
> We thank the reviewer for their suggestions. We would like to emphasize that our work is intended as a foundational investigation into CoT reasoning in LLMs, drawing inspiration from foundational studies such as the Physics of Large Language Models series [e.g., Allen-Zhu et al., 2024; Ye et al., 2024]. Similar to these studies, our aim is to build a rigorous framework that could serve as a basis for future investigations across various reasoning tasks.
>
> While our toy and controlled GSM experiments may involve semi-synthetic settings, we would like to highlight that our arithmetic operations experiments on Llama-3-8b (in Section 5.2) are based on real-world data generated by the Llama-3-8b model without any additional fine-tuning or training. This experiment effectively illustrates that CoT errors in LLMs often involve spurious correlations — for instance, LLMs are prone to failures on mathematical problems with large numbers [Razeghi et al., 2022].
>
> Our experiments demonstrate this phenomenon in real-world datasets, and show that even in the simplest of settings, baseline methods such as ORM and Math Shepherd fail to detect CoT errors accurately, while our information-theoretic approach remains robust and accurate.
> We believe these findings are valuable and significant on their own, as they point toward the possibility of constructing more robust process reward models without relying on annotated CoT datasets. Furthermore, and more importantly, they highlight the limitations of existing baseline methods which rely on the correctness of final answers, as these methods are prone to spurious correlations in model's CoT reasoning, and consequently may not detect the true cause of an error.
>
> While we agree that additional large-scale experiments could further demonstrate the broader applicability of our framework, we view this as an interesting direction for future work.
>
> **References**
>
> Zeyuan Allen-Zhu, Yuanzhi Li. Physics of Language Models: Part 1, Learning Hierarchical Language Structures. 2024
>
> Tian Ye, Zicheng Xu, Yuanzhi Li, Zeyuan Allen-Zhu. Physics of Language Models: Part 2.1, Grade-School Math and the Hidden Reasoning Process. 2024
>
> Tian Ye, Zicheng Xu, Yuanzhi Li, Zeyuan Allen-Zhu. Physics of Language Models: Part 2.2, How to Learn From Mistakes on Grade-School Math Problems. 2024
>
> Yasaman Razeghi, Robert L. Logan IV, Matt Gardner, and Sameer Singh. Impact of pretraining term frequencies on few-shot reasoning, 2022.
>
> > Section 2.1. Do we need to, and how to define the set of lambda in a realistic task?
>
> In practice, solving complex problems often requires breaking them down into sub-operations that a model can directly perform without further decomposition. Our theoretical framework uses tasks (denoted by $\lambda$) to categorize these sub-operations into a finite set of groups (e.g., addition, multiplication), allowing practitioners to evaluate a model's CoT reasoning on any specific sub-task category.
>
> The task of defining $\lambda$ is up to the practitioner and depends on the category they are interested in evaluating. If a practitioner wishes to use our framework to assess a model's performance on sub-operations of a particular type (e.g., addition in the GSM8K dataset or inductive reasoning in the Atlas Reasoning dataset), they would need to choose the relevant category and filter for the sub-operations within this type. They can then calculate the information gain across these filtered steps.
>
> Thus, the categories (or tasks) are domain-dependent, and, in practice, it is only necessary to define the category of interest rather than all possible categories. Alternatively, if the goal is to identify incorrect steps at a sample-wise level within a given CoT trajectory, defining specific categories is unnecessary. This can be achieved by computing sample-wise information gain at each step of the CoT trajectory, as outlined at the end of Section 3 in our paper.
>
> We hope this clarifies the practical definition and use of $\lambda$ in realistic tasks, and we would be happy to discuss further if any questions remain.

---

> > ### Author Response · Authors · 2024-11-18
> > **Rebuttal (continued)**
> >
> > > How do you define a step in a generation task?
> >
> > In our framework, a "step" refers to the sub-tasks that an LLM decomposes a given problem into during its Chain-of-Thought (CoT) reasoning. Each step represents a discrete sub-task or operation that contributes to the overall problem-solving process. For example, in an arithmetic task, steps might include specific operations like addition or multiplication; in reasoning tasks, steps could correspond to deductive or inductive reasoning processes. We will further highlight this in the final version of our paper.
> >
> > We hope this clarifies the definition of a "step" in our context, and we would be glad to discuss further if needed.
> >
> > > It seems like the key is modeling the information gain. However, a separate model approximates the conditional distribution in the proposed framework, which is then used to model the information gain. As we all know, the training of a separate model is determined by the training data and sampled trajectories. How do you guarantee the learned model can model the conditional distribution well? In other words, what is the requirement for training such a model?
> >
> > Thank you for this interesting and thought-provoking question. Recall that the distribution we are estimating is $p(Y \mid X^M_t)$, where $Y$ denotes the ground truth output and $X^M_t$ represents the model’s CoT reasoning up to an intermediate step $t$. In our setup, we assume access to ground truth outputs $Y$ and can obtain the model's CoT reasoning $X^M_t$ by querying the model itself. Thus, we have direct access to the data needed to estimate this conditional distribution. In practice, we use a subset of this data to train the supervisor model.
> >
> > More specifically, as detailed in Appendix B.1.3, we achieve this by fine-tuning another LLM (the supervisor model), which takes the model's CoT reasoning $X^M_t$, and is fine-tuned to predict the ground truth final output $Y$. We use a special token to separate the model's CoT reasoning and the final output during fine-tuning. At inference time, appending this special token to the model input serves as a signal for the supervisor model to predict the final output directly. In this way,$g_{\textup{sup}}(X^M_t)$ approximates the conditional distribution $p(Y \mid X^M_t)$.
> >
> > To ensure the model is trained well, we apply early stopping based on validation losses to avoid overfitting and verify performance using test losses. Finally, to compute the information gain at step $t$, we use a held-out subset of data $\{ (X^M_{t-1}, X^M_t, Y) \}$ and apply the difference defined in Equation (3) in Proposition 3.4 to estimate the information gain.
> >
> > Notably, this fine-tuning process requires only a relatively small supervisor model and a modest amount of fine-tuning data. For example, in our GSM 8K experiments, we used GPT-2 as the supervisor model and fine-tuned it with roughly 2000 samples, which was sufficient to accurately estimate information gain.
> > We hope this clarifies the requirements for training the supervisor model and we will include additional discussion on this in the final version of our paper.
> >
> > > Do we need to guarantee that the training data of the separate model follows a similar distribution of the final task?
> >
> > In our experiments, we used the same distribution of prompts as the original task to generate the model’s CoT reasoning steps, $X^M_t$, for training the supervisor model. This alignment ensures that the supervisor model learns to approximate $p(Y \mid X^M_t)$ accurately under the specific conditions of the target task.
> >
> > However, it is conceivable that training the supervisor model on a similar—but not identical—data distribution could also yield a good estimate of $p(Y \mid X^M_t)$. For instance, if the supervisor model is trained on model generations from one mathematics dataset, it could potentially estimate the information gain for another, similar mathematics dataset reliably.
> > This possibility raises interesting questions about the generalizability of the supervisor model across tasks with related structures, and we consider this an area for future investigation.
> >
> > We hope that we were able to address all the reviewer's questions in the above and hope that the reviewer would consider increasing their score.

---

> > > ### Comment · Reviewer_h9ff · 2024-11-28
> > >
> > > Thanks for the explanation. I agree with the motivation for proposing a theoretical framework and that the proposed concepts are helpful for clearly defined tasks such as math or coding. However, I am still not convinced it could be generalized to more general scenarios with unclear steps. Given that the goal is to propose a foundation theatrical framework, one needs to ensure the framework is generalizable rather than a specific task. Hence, I tend to keep my original rating.

---

> > > > ### Author Response · Authors · 2024-12-01
> > > >
> > > > We thank the reviewer for their comment.
> > > >
> > > > Firstly, we would like to highlight that settings such as mathematical problem solving and coding are important sub-domains in LLM reasoning, and therefore we believe that a principled approach towards accurate CoT evaluations in these settings is a valuable contribution.
> > > >
> > > > Secondly, we believe that our framework could be more generally applied to settings where reasoning steps can be meaningfully categorised into sub-groups of interest. For example, in logical reasoning problems in Common-sense QA, the CoT steps could be categorized into distinct reasoning types, such as causal reasoning (identifying cause-and-effect relationships), temporal reasoning (understanding the sequence and timing of events), or spatial reasoning (comprehending physical arrangements and relationships between objects).
> > > >
> > > > Beyond this, our method could also be applied to settings such as the Blockworlds data. Here, the main task is "planning," which itself can be decomposed into sub-tasks comprising specific sequences or tactics of primitive actions (e.g., "stack," "unstack," "move"). In this setting, our information-gain methodology could be applied to evaluate LLM's performance in planning these sub-tasks. By examining the information gain for each sub-task, we could assess where the reasoning process succeeds or fails.
> > > >
> > > > While these tasks lend themselves well to our framework, we acknowledge that they may introduce additional nuances or challenges, particularly concerning the assumptions underlying our methodology. Exploring such tasks in detail is indeed an exciting direction which we leave for future work.
> > > >
> > > > We hope this addresses the reviewer's concerns regarding our methodology, and will be happy to clarify any other aspects which remain unclear.

---

### Official Review · Reviewer_a8GB · 2024-11-02

**Soundness:** 3
**Presentation:** 3
**Contribution:** 3
**Rating:** 8
**Confidence:** 4

**Summary:**

The paper presents an algebraic framework to model CoT-based reasoning process in an LLM and in the ground truth. Specifically, it is assumed that there is a set of "primitive tasks", the members of which can be used to generate any task via composition. Such a notion of composition also induces a notion of "span" of a set of tasks and the notion of unidenfiability of a task (with respect to a given set of tasks). The key assumptions in this framework is that when the model's CoT for a prompt contains an unidenfiable task with respect to the tasks involved in the CoT of the ground truth, a "fork-structured" Bayesian network is induced, and due to d-separation, certain conditional independence property holds, which in turn suggests that after the forking point, information gain becomes zero.

Using this framework, the paper introduce a new scheme for evaluating the CoT of LLM. Experimental results are given demonstrating the advantage of this framework and the proposed scheme.

**Strengths:**

The algebraic formulation and information-theoretic perspective in this paper are very nice. The approach taken is clean. Although the assumptions (particularly Assumption 3.2) may be subject to debate, I think this work is making an important step towards theoretical understanding of the CoT in LLM.  The presentation is also very good. I very much enjoy reading this paper.

**Weaknesses:**

I have only one concern about this paper, namely, I question if the sample-wise information gain (Equation 5) is a valid measure.

Note that although the standard mutual information (or "distribution-wise mutual information") is well-known to be non-negative and zero mutual information gives independence, the "sample-wise mutual information" (namely, the term without expectation) can be negative in general. The "intuitive" argument the authors give (lines 299 to 303)  to justify the use of sample-wise mutual information is not convincing to me. -- I feel this is the main weakness of an otherwise very elegant development. I encourage the authors to resolve this issue.

**Questions:**

See weakness.

---

> ### Author Response · Authors · 2024-11-18
> **Rebuttal**
>
> Firstly, we sincerely thank the reviewer for their thoughtful and positive feedback. We greatly appreciate their recognition of our framework's novelty, theoretical contributions, and the clarity of our presentation. Below, we address the concern raised regarding sample-wise information gain.
>
> > I have only one concern about this paper, namely, I question if the sample-wise information gain (Equation 5) is a valid measure.
> Note that although the standard mutual information (or "distribution-wise mutual information") is well-known to be non-negative and zero mutual information gives independence, the "sample-wise mutual information" (namely, the term without expectation) can be negative in general. The "intuitive" argument the authors give (lines 299 to 303) to justify the use of sample-wise mutual information is not convincing to me. -- I feel this is the main weakness of an otherwise very elegant development. I encourage the authors to resolve this issue.
>
> Thank you for highlighting this point and we appreciate the opportunity to clarify our sample-wise information gain approach.
> 1. Sample-Wise Information Gain as a Heuristic Measure
>
> We propose sample-wise information gain as a heuristic measure to assess sub-task correctness at a sample-specific level, intended as a signal to detect incorrect steps within a given CoT sample. While it is true that sample-wise information gain can be negative, this quantity intuitively reflects the change in the model's confidence about the ground truth answer $Y$ being correct. If the sample-wise information gain is positive, it indicates an increase in the model’s confidence about the correct answer (i.e. $p(Y\mid X^M_t) > p(Y\mid X^M_{t-1})$) , suggesting that the model is proceeding along the correct reasoning path. Conversely, if the sample-wise information gain is negative, it indicates a decrease in confidence about the correct answer, signalling a potential error in the reasoning step.
>
> 2. Precedent in Prior Work
>
> While this approach does not test the conditional independence in Theorem 3.3 in our paper, there is precedent for using sample-wise mutual information in prior work to make inferences at the individual sample level. For instance, Durme et al., (2009), Ethayarajh et al. (2022), Nandwani et al., (2023) leverage sample-wise mutual information for making inferences on samples. Inspired by these methods, we leverage sample-wise information gain as a practical heuristic to evaluate reasoning steps within individual CoT trajectories.
>
> 3. Empirical Evidence Supporting Sample-Wise Information Gain
>
> Importantly, our experimental results (Tables 1–3) demonstrate that sample-wise information gain effectively captures information about the correctness of CoT steps. These results show that this metric can detect CoT errors with higher accuracy than the baselines considered, underscoring its practical utility in real-world settings. While we agree that further theoretical grounding would strengthen its justification, empirical evidence supports its efficacy in detecting reasoning errors.
> We hope this explanation addresses your concern and highlights the rationale behind the use of sample-wise information gain, and we will further clarify this in the final version of our paper.
>
>  Thank you again for your thoughtful feedback and for recognizing the broader contributions of our framework.
>
> **References**
>
> Benjamin Durme, Ashwin Lall. Streaming Pointwise Mutual Information, 2009.
>
> Kawin Ethayarajh, Yejin Choi, Swabha Swayamdipta. Understanding Dataset Difficulty with V-Usable Information, 2022.
>
> Yatin Nandwani, Vineet Kumar, Dinesh Raghu, Sachindra Joshi, Luis A. Lastras. Pointwise Mutual Information Based Metric and Decoding Strategy for Faithful Generation in Document Grounded Dialogs, 2023.

---

> > ### Comment · Reviewer_a8GB · 2024-11-21
> > **Thanks for the explanation.**
> >
> > I guess what I was hoping for is beyond intuitive arguments, experimental evidence, or that so and so have done the same.  Although to me this is a loose end, I think the paper is good enough to be accepted.
> >
> > I will keep my score.

---

### Official Review · Reviewer_a3yv · 2024-11-03

**Soundness:** 2
**Presentation:** 2
**Contribution:** 2
**Rating:** 5
**Confidence:** 4

**Summary:**

The paper presents an information-theoretic framework for evaluating the quality of Chain-of-Thought (CoT) reasoning in Large Language Models (LLMs). The authors propose a method that quantifies the "information gain" at each reasoning step to identify failure modes without requiring expensive annotated datasets. The research shows that this approach is more effective than existing methods such as Outcome Reward Modeling (ORM) and Math-Shepherd in assessing intermediate steps in reasoning, especially when dealing with complex CoT processes. The authors claim that with the presented framework, intermediate steps can be used to detect errors, rather than just relying on the final answer. This is a step towards more interpretable CoT reasoning.

**Strengths:**

Applying an information theory perspective to the CoT reasoning chain provides a novel and rigorous method for evaluating the contribution of each step to the final output. This fresh approach differs from existing methods by quantifying the information gain at each reasoning step, rather than relying only on the final answer. This makes it possible to identify when errors occur, improving the interpretability and reliability of the model.

Unlike other methods, it does not require annotated data to train a verifier model, providing a cost-effective alternative to process-monitoring style models. By eliminating this requirement, it provides a scalable solution that is particularly useful for domains or tasks where generating annotated CoT data is impractical or prohibitively expensive.

Finally, the authors experimented with a toy dataset and a real dataset (GSM8K) and compared their approach to baselines. The improvements over the baseline demonstrate the effectiveness of the method.

**Weaknesses:**

I am still unsure about the experimental design, which contradicts some claims of the paper. For example, the use of a supervising model requires a lot of data, which is needed to train the supervising model. This requirement adds computational complexity, requiring additional processing power, memory, and time that the authors claim is not needed compared to the process monitoring models in the introduction.

In addition, using a supervising model to predict the answer given the previous steps in the chain of reasoning as a way to understand whether the previous steps contain information needed to answer a question may not be correct. For example, many questions can be answered directly by the model in an answer-only setup ("The answer is X"). For these questions, no CoT reasoning is needed. But in this framework this will not be the case and these steps will be listed as important steps. Basically, the framework's sample-wise information gain is designed to detect errors on a per-sample basis. However, there are instances where a model may accidentally reach the correct final answer, even with incorrect intermediate steps. This can lead to "false negatives," where a faulty chain of reasoning is not flagged because the final answer happens to be correct.

Similarly, a lot of time CoT reasoning can hurt performance if the initial step is wrong and then it is hard for the model to recover (https://arxiv.org/abs/2311.07945). In such cases, the steps are considered important if the model can predict the answer correctly, which in many cases is not the case if the start is wrong. There is no analysis done in the paper.
Also, the advantage of the approach presented in the paper is to understand and find the errors in the intermediate steps compared to predicting the final answer. This analysis is missing in the paper, especially in the areas where the intermediate errors are analyzed in detail.

The experimental results rely heavily on controlled settings, such as toy datasets and GSM-8K, where task types and error points are clearly defined. In real-world scenarios, errors in CoT reasoning may not follow the structured patterns present in these datasets. This reliance on controlled experiments raises the question of how well the framework would perform on naturally occurring data with more diverse error types. This is important for understanding the scalability of the method as the number of reasoning steps increases or the chain becomes more complicated.

**Questions:**

There needs to be an analysis of questions where the model can directly predict the answers and how this framework addresses those questions. Are intermediate steps needed in such cases? And what about false negatives, where a faulty chain of reasoning is not flagged because the final answer happens to be correct? This analysis needs to be presented in the paper.

The analysis of errors made at different steps needs more analysis. Even better if these stepwise errors form some patterns. How reliable the supervisor model is in identifying these errors would be interesting to see.
Cost comparison between training the supervisor model presented in the paper and the process supervision paper needs to be analyzed to determine if training a supervisor model is even cheaper as claimed in the paper.

Doubts about theoretical assumptions: Does the framework assume a linear sequence of reasoning steps, and how would it handle nonlinear or branching reasoning paths? Or what about cases where multiple reasoning chains lead to the correct answer?

---

> ### Author Response · Authors · 2024-11-18
> **Rebuttal**
>
> First of all, we thank the reviewer for their comprehensive review to improve our paper. In particular, we appreciate the reviewer acknowledging that our method "provides a novel and rigorous method for evaluating the contribution of each step to the final output". Below, we take the opportunity to clarify some of the concerns and misunderstandings stated in the above review.
>
> > I am still unsure about the experimental design, which contradicts some claims of the paper. For example, the use of a supervising model requires a lot of data, which is needed to train the supervising model. This requirement adds computational complexity, requiring additional processing power, memory, and time that the authors claim is not needed compared to the process monitoring models in the introduction.
>
> We believe that there might be a misunderstanding in terms of the comparisons we made in the introduction [lines 38-40]. In particular, the introduction aimed to highlight that existing process reward models required human annotations on the correctness of each of the steps as well as training the subsequent reward model. In our case, we aim to remove the burden of expensive human annotations, while the training requirement remains. We apologize for this misunderstanding and will ensure that this aspect is clarified in the final version of this paper.
>
> > In addition, using a supervising model to predict the answer given the previous steps in the chain of reasoning as a way to understand whether the previous steps contain information needed to answer a question may not be correct. For example, many questions can be answered directly by the model in an answer-only setup ("The answer is X"). For these questions, no CoT reasoning is needed. But in this framework, this will not be the case and these steps will be listed as important steps.
> There needs to be an analysis of questions where the model can directly predict the answers and how this framework addresses those questions. Are intermediate steps needed in such cases?
>
> We thank the reviewer for bringing up this interesting aspect of "answer-only" setup in our framework, i.e. "what is 2+2?". However, we believe that there might be some confusion here. If a question indeed does not require CoT, then there will be exactly a single step in our framework (Prompt -> Final Answer). In this setting, our framework still works (by checking if the predicted answer matched the ground truth answer/label) but would be considered a trivial case. Hence, this paper and our framework primarily focuses on the CoT settings, where multiple steps are required and hence each step adds information if correct.
>
> Could the reviewer please elaborate on their statement "But in this framework this will not be the case and these steps will be listed as important steps.". Given that there are no intermediate steps, our framework will not introduce new steps and hence will also not list them as important. We are happy to discuss further if this point remains unanswered.

---

> ### Author Response · Authors · 2024-11-18
> **Rebuttal (continued)**
>
> >Basically, the framework's sample-wise information gain is designed to detect errors on a per-sample basis. However, there are instances where a model may accidentally reach the correct final answer, even with incorrect intermediate steps. This can lead to "false negatives," where a faulty chain of reasoning is not flagged because the final answer happens to be correct.
> And what about false negatives, where a faulty chain of reasoning is not flagged because the final answer happens to be correct? This analysis needs to be presented in the paper.
>
> Next, we would like to highlight that, our proposed method is primarily focused on a mutual information level (average) but can also be applied to a "per-sample basis", similar to V-usable information paper [Ethayarajh et al., 2022].
> However, we thank the reviewer for pointing out this special case. We would like to clarify a few things regarding the setup where the LLMs get the correct final answer despite incorrect intermediate steps.
>
> Recall that our method only estimates the relevant information for the correct final answer added by each successive step. Therefore, in cases where an intermediate step is incorrect, it should add no useful information regarding the correct final answer and therefore the information gain at that step should still be 0 (regardless of whether the final output is correct or not).
> In contrast, in cases where the model indeed consistently commits errors at intermediate steps which then consistently lead to the correct answer, the baselines such as ORM and Math Shepherd would fail to identify such errors as these baselines mainly rely on the correctness of the final answer.
>
> Nevertheless, your question did spark some interest in us and hence we went back to our datasets to check how often this indeed happens and report that in our realistic arithmetic experiments in Section 5.2 only 1.2% of the samples indeed exhibit this behaviour where an intermediate step is wrong, but the final answer is correct. For the synthetic toy experiment, this never happens as we have control over the generation to some degree.
>
> Hence, we would argue that for our settings, these are relatively rare and would not affect our framework. We thank the reviewer again for their insightful comment to improve our paper and will definitely add a discussion on this in the final version of our paper.
>
> **Reference:**
> Kawin Ethayarajh, Yejin Choi, Swabha Swayamdipta. Understanding Dataset Difficulty with V-Usable Information, 2022.
>
> > Similarly, a lot of time CoT reasoning can hurt performance if the initial step is wrong and then it is hard for the model to recover (https://arxiv.org/abs/2311.07945). In such cases, the steps are considered important if the model can predict the answer correctly, which in many cases is not the case if the start is wrong. There is no analysis done in the paper. Also, the advantage of the approach presented in the paper is to understand and find the errors in the intermediate steps compared to predicting the final answer. This analysis is missing in the paper, especially in the areas where the intermediate errors are analyzed in detail.
>
> The reviewer raises an interesting point, which is that of: "What happens if the model fails to get the initial steps right?". Here, we strongly believe that there is a misunderstanding regarding our experimental setup as we have indeed performed a comprehensive ablation study where we consider settings with errors occurring at each possible step (see Figure 3).
>
> For example, in our toy experiment setting, we have a setup, where we are able to control all 5 subtasks which are all necessary to complete the task. One by one, we then analyse the behaviour of information gain as each of the 5 subtasks becomes "unidentifiable"/wrong. Specifically, $LLM_1$ in our toy experiments (Section 5.1) corresponds to the setting where the initial step of CoT reasoning is wrong while the subsequent tasks are executed correctly.
>
> In the experimental results shown in Figure 3, we observe that information gain becomes negligible (or even negative) as soon as we encounter a wrong step. Additionally, this information gain at subsequent steps remains zero even when the subsequent steps are performed correctly, which is also consistent with our theoretical framework for identifiability outlined in Section 3.

---

> > ### Author Response · Authors · 2024-11-18
> > **Rebuttal (continued)**
> >
> > > The experimental results rely heavily on controlled settings, such as toy datasets and GSM-8K, where task types and error points are clearly defined. In real-world scenarios, errors in CoT reasoning may not follow the structured patterns present in these datasets. This reliance on controlled experiments raises the question of how well the framework would perform on naturally occurring data with more diverse error types. This is important for understanding the scalability of the method as the number of reasoning steps increases or the chain becomes more complicated.
> >
> > We thank the reviewer for their suggestions. We would like to emphasize that our work is intended as a foundational investigation into CoT reasoning in LLMs, drawing inspiration from foundational studies such as the Physics of Large Language Models series [e.g., Allen-Zhu et al., 2024; Ye et al., 2024]. Similar to these studies, our aim is to build a rigorous framework that could serve as a basis for future investigations across various reasoning tasks.
> >
> > While our toy and controlled GSM experiments may involve semi-synthetic settings, we would like to highlight that our arithmetic operations experiments on Llama-3-8b (in Section 5.2) are based on real-world data generated by the Llama-3-8b model without any additional fine-tuning or training. This experiment effectively illustrates that CoT errors in LLMs often involve spurious correlations — for instance, LLMs are prone to failures on mathematical problems with large numbers [Razeghi et al., 2022]. Our experiments demonstrate this phenomenon in real-world datasets, and show that even in the simplest of settings, baseline methods such as ORM and Math Shepherd fail to detect CoT errors accurately, while our information-theoretic approach remains robust and accurate.
> >
> > We believe these findings are valuable and significant on their own, as they point toward the possibility of constructing more robust process reward models without relying on annotated CoT datasets. Furthermore, and more importantly, they highlight the limitations of existing baseline methods which rely on the correctness of final answers, as these methods are prone to spurious correlations in model's CoT reasoning, and consequently may not detect the true cause of an error.
> > While we agree that additional large-scale experiments could further demonstrate the broader applicability of our framework, we view this as an interesting direction for future work.
> >
> > **References**
> >
> > Zeyuan Allen-Zhu, Yuanzhi Li. Physics of Language Models: Part 1, Learning Hierarchical Language Structures. 2024
> >
> > Tian Ye, Zicheng Xu, Yuanzhi Li, Zeyuan Allen-Zhu. Physics of Language Models: Part 2.1, Grade-School Math and the Hidden Reasoning Process. 2024
> >
> > Tian Ye, Zicheng Xu, Yuanzhi Li, Zeyuan Allen-Zhu. Physics of Language Models: Part 2.2, How to Learn From Mistakes on Grade-School Math Problems. 2024
> >
> > Yasaman Razeghi, Robert L. Logan IV, Matt Gardner, and Sameer Singh. Impact of pretraining term frequencies on few-shot reasoning, 2022.
> >
> >
> > > The analysis of errors made at different steps needs more analysis. Even better if these stepwise errors form some patterns. How reliable the supervisor model is in identifying these errors would be interesting to see.
> >
> > We thank the reviewer for this indeed very thought-provoking comment on the case where the stepwise errors form some patterns. In fact, our experiments already cover these cases to some extent. For example, in our synthetic experiment, where we have 5 sub-tasks and are able to control which ones are correctly performed and which ones are not.
> >
> > More specifically, in our synthetic data experiment for $LLM_3$, the failure or success of the third step is dependent on the output at step 2, (i.e., step 3 fails if only if $z_2[2] >150$) and hence forms a pattern. Similar patterns exist in our arithmetic operations experiment in Section 5.2, where we observe that the addition fails mostly when exactly one of the variables $(x, y)$ is large and the other is small (as evidenced by Figure 4). In all of these experiments, our proposed method is still able to reliably identify the errors which cannot be said of existing methods such as ORM or Math Shepard.
> >
> > In fact, in cases where such patterns arise, our experimental results in Section 5 show that the baselines ORM or Math Shepherd often capture spurious correlations arising from these patterns, thereby leading to misleading conclusions, whereas our method accurately flags the incorrect step even in the presence of these spurious correlations.
> >
> > We hope that this addresses the reviewer's concerns regarding patterns arising in stepwise errors and will be happy to discuss further if any aspect remains unaddressed. We will make sure to highlight this comment in the final version of the paper.

---

> > > ### Author Response · Authors · 2024-11-18
> > > **Rebuttal (final)**
> > >
> > > > Cost comparison between training the supervisor model presented in the paper and the process supervision paper needs to be analyzed to determine if training a supervisor model is even cheaper as claimed in the paper.
> > >
> > > We believe that there is a misunderstanding in terms of what we have claimed. We claim that collecting human-annotated data on the correctness of each subtask is expensive. We do not claim that the training of the corresponding reward model or supervisor models is cheaper. We are merely stating that current process reward models require expensive human-annotated data, which our method is able to circumvent.
> > >
> > > > Doubts about theoretical assumptions: Does the framework assume a linear sequence of reasoning steps, and how would it handle nonlinear or branching reasoning paths? Or what about cases where multiple reasoning chains lead to the correct answer?
> > >
> > > Our framework follows the standard Bayesian network assumptions and does not impose any linearity between the nodes. Regarding the mention of "nonlinear or branching reasoning paths," we kindly ask the reviewer to provide further clarification on what they mean by "non-linear reasoning paths." This would help us better understand the specific concern or question being raised and address it more precisely.
> > >
> > > We also thank the reviewer for raising the interesting question about multiple reasoning chains leading to the correct answer, which we also briefly hinted at on line 142. To address this scenario, our framework inherently accommodates multiple reasoning chains, as it does not assume the existence of a unique decomposition for any given task (as noted on line 142). As long as the sub-tasks in the reasoning process are correct and contribute to progress toward the final answer, there will be measurable "information gain" regardless of the specific path taken. Conversely, when the LLM performs incorrect sub-tasks that lead away from the correct answer, the information gain will become null.
> > > We greatly appreciate the reviewer’s insightful question and will make sure to clarify this aspect in the revised manuscript.
> > >
> > > We hope the above clarifies some of the concerns and misunderstandings regarding our work and that the reviewer will consider increasing their score. We will be happy to further clarify any questions that remain unanswered.

---

> ### Comment · Reviewer_a3yv · 2024-11-25
> **Reponse**
>
> >We thank the reviewer for bringing up this interesting aspect of "answer-only" setup in our framework, i.e. "what is 2+2?".
>
> Thanks for the response. I am not talking about the cases where the solution can be reached in one step like the example question presented above.
>
> Rather I am talking about a MWP like "John has 10 apples, John gave 5 apples to Jack, ...?". In such cases, CoT is a good choice to solve with the intermediate steps. But the model can also directly say the answer in one step. What about those cases?  And also those cases where CoT reasoning is incorrect but the final answer is correct?
>
> I understand that the framework will not introduce extra steps but an analysis is needed around "false negatives," where a faulty chain of reasoning is not flagged because the final answer happens to be correct. In this specific dataset, it is 1.2% as the authors say (cases when reasoning is incorrect but the final ans is correct) but in general it is a problem as a lot of models memorize the answers.

---

> > ### Author Response · Authors · 2024-11-29
> > **Thanks for your response**
> >
> > We thank the reviewer for their comment. We believe there are still some fundamental misunderstandings regarding our methodology which we hope to clarify below:
> > > Rather I am talking about a MWP like "John has 10 apples, John gave 5 apples to Jack, ...?". In such cases, CoT is a good choice to solve with the intermediate steps. But the model can also directly say the answer in one step. What about those cases?
> >
> > The goal of our framework is to detect erroneous CoT steps, regardless of whether a CoT is strictly necessary for solving a given problem. In cases where a model can directly produce the final answer but is instead queried to think step-by-step (using CoT reasoning), our method evaluates the correctness of each CoT step independently. Specifically, if a CoT step is correct, the information gain should be positive, while for an incorrect step, the information gain should be zero. This ensures that our framework can effectively detect CoT errors even in scenarios where intermediate reasoning steps are not necessarily needed.
> >
> > > And also those cases where CoT reasoning is incorrect but the final answer is correct?
> > I understand that the framework will not introduce extra steps but an analysis is needed around "false negatives," where a faulty chain of reasoning is not flagged because the final answer happens to be correct. In this specific dataset, it is 1.2% as the authors say (cases when reasoning is incorrect but the final ans is correct) but in general it is a problem as a lot of models memorize the answers.
> >
> > We would like to clarify that in our methodology, the information gain at an intermediate step is **independent** of the model's final output (because the information gain only depends on the model's CoT reasoning **up to** the intermediate step of interest). This ensures that when an intermediate CoT step is incorrect, the information gain at that step will be zero, **regardless of whether the final answer is correct or not**. For instance, in our toy experiments, if the model arrived at the correct final answer despite having incorrect CoT steps, the information gain for intermediate steps will remain unchanged, allowing our method to flag the faulty or irrelevant intermediate steps.
> >
> > This is, in fact, a strength of our approach and a key distinction from outcome-based baselines such as ORM or Math Shepherd. These baselines rely on the correctness of the final model answer, which can lead to false negatives in cases where the final answer is consistently correct despite errors in intermediate reasoning steps. This is also evident from our empirical results on the controlled GSM-8K data shown in the table below, which shows that false negative rate (FNR) of our method is considerably lower than that of the ORM baseline:
> >
> > | Method | FNR $\downarrow$ |
> > | -------- | ------- |
> > | IG (Ours) |  **0.02** |
> > | ORM | 0.13 |
> >
> > Therefore, our method avoids pitfalls associated with checking the correctness of the final model answer by instead focusing on the correctness of each CoT step through an information-theoretic perspective.
> >
> > We hope the above clarifies any misunderstandings regarding our methodology and its motivations. We will be happy to address any aspects which remain unclear.

---

> > > ### Author Response · Authors · 2024-12-02
> > >
> > > With the rebuttal period coming to an end, we would like to thank the reviewer again for their suggestions and feedback to improve our paper.
> > >
> > > As per the reviewer's request, we have included the false negative rate (FNR) from our method (IG) as well as the ORM baseline in the results above. Additionally, we have also included the FNR for an additional experiment on PRM800K in our general comment above. Both of these results show that our method's FNR is lower than that of ORM. We hope that these additional results provided above address the reviewer's concerns.
> > >
> > > If there are any outstanding questions or concerns, we would be more than happy to address them over the remaining rebuttal period. Otherwise, we hope that the reviewer would consider increasing their score.

---

### Official Review · Reviewer_qEak · 2024-11-04

**Soundness:** 3
**Presentation:** 3
**Contribution:** 3
**Rating:** 6
**Confidence:** 4

**Summary:**

The paper proposes to use information theory to quantify the information gain of each reasoning step in a COT solution. The notation of task is introduced for each reasoning step, and assumptions are made for reasoning errors being unidentifiable primitive tasks by the LLMs. The information gain is formulated as a conditional probability difference between two subsequent steps with respect to the final correct answer. A supervisor model is trained to predict such conditional probability. A toy data with 5 primitive tasks are constructed to train the supervisor model and to test the efficacy of the proposed method against baselines including ORM and Math-Shepherd. The paper demonstrated the effectiveness of the Information Gain method to capture errors in reasoning steps (Figure 3 and Table 1).

**Strengths:**

1. The idea of using information gain to detect the helpfulness of each reasoning step is very novel and interesting.
2. The controlled experiments using toy data and GSM8K are well conducted and clearly demonstrate the advantage of Information Gain over baseline methods such as ORM.
3. The paper is well written and easy to follow.

**Weaknesses:**

1. A significant weakness of the paper is that it is unclear how applicable it is to real world reasoning tasks. More experiments involving more challenging tasks such as MMLU or MATH would be more convincing.
2. The assumption of needing to annotate the task involved in each step is too restrictive. In reality, such a one-to-one mapping is neither possible nor necessary. A single reasoning step could use more than a single primitive task. It could also use no primitive task at all by simply being a filer step.
3. The distinction between error steps and irrelevant steps is mentioned but it is unclear how the IG method handles them separately. More clarity is needed. From the description, it looks like they will be treated equally. However, irrelevant steps are not as harmful as error steps.

**Questions:**

1. The paper can benefit from grammar checks.
Line 055 `detect` is repeated twice.
Line 164 `tasks` is supposed to be `task`.
Line 295, `which step went wrong` is using past tense while most of the paper is using present tense. Same for `chose` in line 429.
2. The proposition 3.4 is problematic as Equation (3) is asymmetric with respect to Y and Xj, while we know that mutual information should be symmetric.

---

> ### Author Response · Authors · 2024-11-18
> **Rebuttal**
>
> First of all, we would like to thank the reviewer for their time and thoughtful comments on our paper. We greatly appreciate the positive feedback on the novelty of using information gain for detecting reasoning errors, the clarity of our presentation, and the robustness of our experiments. Below we address the questions raised by the reviewer:
>
> > A significant weakness of the paper is that it is unclear how applicable it is to real-world reasoning tasks. More experiments involving more challenging tasks such as MMLU or MATH would be more convincing.
>
> We thank the reviewer for their suggestions. We would like to emphasize that our work is intended as a foundational investigation into CoT reasoning in LLMs, drawing inspiration from foundational studies such as the "Physics of Large Language Models series" [e.g., Allen-Zhu et al., 2024; Ye et al., 2024]. Similar to these studies, our aim is to build a rigorous framework that could serve as a basis for future investigations across various reasoning tasks.
>
> While our toy and controlled GSM experiments may involve semi-synthetic settings, we would like to highlight that our arithmetic operations experiments on Llama-3-8b (in Section 5.2) are based on real-world data generated by the Llama-3-8b model without any additional fine-tuning or training. This experiment effectively illustrates that CoT errors in LLMs often involve spurious correlations — for instance, LLMs are prone to failures on mathematical problems with large numbers [Razeghi et al., 2022]. Our experiments demonstrate this phenomenon in real-world datasets and show that even in the simplest of settings, baseline methods such as ORM and Math Shepherd fail to detect CoT errors accurately, while our information-theoretic approach remains robust and accurate.
> We believe these findings are valuable and significant on their own, as they indicate the possibility of constructing more robust process reward models without relying on annotated CoT datasets.
>
> Furthermore, and more importantly, they highlight the limitations of existing baseline methods which rely on the correctness of final answers, as these methods are prone to spurious correlations in model's CoT reasoning, and consequently may not detect the true cause of an error.
> While we agree that additional large-scale experiments could further demonstrate the broader applicability of our framework, we view this as an interesting direction for future work.
>
>
> **References**
>
> Zeyuan Allen-Zhu, Yuanzhi Li. Physics of Language Models: Part 1, Learning Hierarchical Language Structures. 2024
>
> Tian Ye, Zicheng Xu, Yuanzhi Li, Zeyuan Allen-Zhu. Physics of Language Models: Part 2.1, Grade-School Math and the Hidden Reasoning Process. 2024
>
> Tian Ye, Zicheng Xu, Yuanzhi Li, Zeyuan Allen-Zhu. Physics of Language Models: Part 2.2, How to Learn From Mistakes on Grade-School Math Problems. 2024
>
> Yasaman Razeghi, Robert L. Logan IV, Matt Gardner, and Sameer Singh. Impact of pretraining term frequencies on few-shot reasoning, 2022.

---

> > ### Comment · Reviewer_qEak · 2024-11-21
> >
> > I disagree that we can call the data used in Section 5.2 "real-world" data. It is still synthetic data generated with arithmetic operations. Furthermore, this paper proposes a new method of information theory to detect CoT errors. In my opinion, this is a not a framework shift like what the authors claimed. If the authors really want to convince the readers the method is generalizable, I think experiments on existing hard reasoning benchmarks like MATH or MMLU are necessary.

---

> ### Author Response · Authors · 2024-11-18
> **Rebuttal (continued)**
>
> >The assumption of needing to annotate the task involved in each step is too restrictive. In reality, such a one-to-one mapping is neither possible nor necessary. A single reasoning step could use more than a single primitive task. It could also use no primitive task at all by simply being a filer step.
>
> We would like to clarify that in our approach, the goal is to evaluate the model's CoT performance on sub-tasks of different kinds. To this end, in practice, our framework uses the notion of primitive tasks to define distinct categories for each sub-step, allowing the model's performance on each category to be evaluated.
>
> For example, in the GSM8K dataset, these categories could include basic arithmetic operations such as addition or multiplication. In reasoning datasets (e.g., Atlas-Reasoning, commonsenseQA), categories could correspond to inductive reasoning, deductive reasoning, or quantitative reasoning. The categories are distinct enough that a suitable CoT decomposition should not combine multiple categories into a single sub-task. If such a combination arises, it suggests that the sub-task could be further decomposed into steps aligned with each individual category.
>
> In this case, our framework allows prompting the model to decompose problems in a way that ensures each sub-step aligns with at most one category. For example, when prompting the LLM on gsm-8k, we explicitly instructed the model to ensure that each step should correspond to distinct sub-operations such as addition, multiplication, etc. This approach preserves interpretability and the granularity needed for reliable CoT performance evaluation, even in complex reasoning tasks.
>
> Moreover, our framework does not require a strict one-to-one mapping of each reasoning step to a specific category. Instead, we need each CoT step to correspond to at most one category and our method remains applicable in cases where a step is simply a filler step and therefore does not belong to any category. In such cases, while the information gain for the filler step may be 0, the information gain for other categories should still convey the same information about model performance.
>
>
> > The distinction between error steps and irrelevant steps is mentioned but it is unclear how the IG method handles them separately. More clarity is needed. From the description, it looks like they will be treated equally. However, irrelevant steps are not as harmful as error steps.
>
> Thank you for raising this important point. Generally speaking, in our framework it is indeed possible to distinguish between irrelevant steps and error steps through the Information Gain (IG) metric. If a step is irrelevant, the information gain at that step may be zero, but as long as the model’s CoT reasoning remains on track toward the correct solution, the information gain will increase at future steps. In contrast, if a step is incorrect and causes the model’s CoT to deviate from the correct path, the information gain will remain zero in all subsequent steps.
>
> Thus, generally by analyzing the information gain in future steps, our framework can effectively differentiate between steps that are irrelevant and those that are genuinely erroneous, capturing their differing impacts on the model's reasoning path.
>
>
>
> > The paper can benefit from grammar checks. Line 055 detect is repeated twice. Line 164 tasks is supposed to be task. Line 295, which step went wrong is using past tense while most of the paper is using present tense. Same for chose in line 429.
>
> We thank the reviewer for pointing out these typos. We will correct them in the final version of our manuscript.
>
> > The proposition 3.4 is problematic as Equation (3) is asymmetric with respect to Y and Xj, while we know that mutual information should be symmetric.
>
> We thank the reviewer for pointing out this potential source of confusion. We would like to clarify that the conditional mutual information in Equation (3), $\mathcal{I}(Y; X^M_j \mid X^M_{j-1})$, is indeed symmetric with respect to $Y$ and $X^M_j$. Specifically, this conditional mutual information is defined as
> $$\mathcal{I}(Y; X^M_j \mid X^M_{j-1}) = E\left[\log \frac{p(Y, X^M_j \mid X^M_{j-1})}{p(Y \mid X^M_{j-1}) \, p(X^M_j \mid X^M_{j-1})}\right],$$
> where both the numerator and the denominator in the expectation above (as well as the expectation itself) are symmetric with respect to $Y$ and $X^M_j$.
> This notation for conditional mutual information follows standard conventions in literature [Wyner (1978), Fawzi et al., (2015)], and we will clarify this further in the final version of our paper.
>
>
>
> We hope our clarifications have addressed the reviewer's concerns, and the reviewer will consider increasing their score.
>
>
>
>
> **Reference:**
>
> A.D. Wyner, A definition of conditional mutual information for arbitrary ensembles, Information and Control, Volume 38, Issue 1, 1978, Pages 51-59
>
> Omar Fawzi, Renato Renner, Quantum conditional mutual information and approximate Markov chains. 2015

---

> > ### Comment · Reviewer_qEak · 2024-11-21
> >
> > While the formula the authors wrote down in the reply is symmetric, the formula in the first half of Eq.(3) is not symmetric wrt to Y and X_j. So, it is confusing to me how the authors arrive at Eq.(3), which is the foundation of the entire method.

---

> > > ### Author Response · Authors · 2024-11-22
> > > **Reply to Reviewer qEak**
> > >
> > > Firstly, we thank the reviewer for their continued engagement and thoughtful feedback.
> > >
> > > > I disagree that we can call the data used in Section 5.2 "real-world" data. It is still synthetic data generated with arithmetic operations. Furthermore, this paper proposes a new method of information theory to detect CoT errors. In my opinion, this is a not a framework shift like what the authors claimed. If the authors really want to convince the readers the method is generalizable, I think experiments on existing hard reasoning benchmarks like MATH or MMLU are necessary.
> > >
> > > Regarding the experiments, we would like to reiterate that our primary focus in this paper is to demonstrate that, even in simple settings (such as our arithmetic problems or controlled GSM/Toy experiments), outcome-based methods like ORMs and Math Shepherd, are prone to spurious correlations and fail to accurately identify CoT errors.
> > >
> > > Our intention, in line with foundational works such as the Physics of Large Language Models series (e.g., Allen-Zhu et al., 2024; Ye et al., 2024), is to highlight to the research community that these limitations of outcome-based methods are already present even in relatively simple settings. These challenges require attention because they can lead to incorrect conclusions when evaluating CoT reasoning. Thus, while we agree that applying our method to challenging tasks like MATH and MMLU would be valuable, we view this as an important direction for future work rather than the focus of this foundational study.
> > >
> > > Our proposed framework avoids the pitfalls of spurious correlations associated with the baselines by adopting an information-theoretic perspective that evaluates how much additional information each step contributes, rather than relying solely on the correctness of the final answer.
> > >
> > > We greatly appreciate the reviewer’s feedback and will ensure that our claims are presented appropriately in the final version of the paper to reflect this scope and intention.

---

> > > > ### Comment · Reviewer_qEak · 2024-11-24
> > > >
> > > > To me, it is unclear
> > > > 1) whether the current method can apply to real-world benchmarks such as MATH and MMLU because of all the assumptions and restrictions on the toy tasks.
> > > > 2) if the answer to 1) is yes, does it still bring benefits to the results?
> > > >
> > > > The authors keep claiming their work to be foundational, but there are way too many so-called foundational methods that are useless practically. My review experience so far is that the authors are very reluctant to do any additional work to increase the impact and applicability of the work. Though I like the novel idea of the work, but I am not fully convinced the method goes beyond the toy tasks and the restrictions. I will keep my score.

---

> > > > > ### Author Response · Authors · 2024-12-01
> > > > > **Additional experimental results**
> > > > >
> > > > > We thank the reviewer again for their continued engagement.
> > > > >
> > > > > Following the reviewer's suggestion regarding experiments on the MATH dataset, we have over the last few days been running additional experiments on OpenAI's PRM800K dataset [Lightman et al., 2023] (which is obtained by labelling the intermediate CoT steps of the MATH dataset). The results for these experiments which have been detailed in our general comment above, show that our method remains applicable on this data, and provides a better over-all accuracy than the ORM baseline.
> > > > >
> > > > > We hope that these experiments address the concerns raised by the reviewer and we will include these in the final version of our paper.
> > > > >
> > > > > **References**
> > > > >
> > > > > Hunter Lightman, Vineet Kosaraju, Yura Burda, Harri Edwards, Bowen Baker, Teddy Lee, Jan Leike, John Schulman, Ilya Sutskever, Karl Cobbe. Let's Verify Step by Step. 2023

---

> > > > > > ### Author Response · Authors · 2024-12-02
> > > > > >
> > > > > > With the rebuttal period coming to an end, we would like to thank the reviewer again for their suggestions and feedback to improve our paper.
> > > > > >
> > > > > >
> > > > > > We hope that our additional experimental results provided above address the reviewer's concerns. If there are any outstanding questions or concerns, we would be more than happy to address them over the remaining rebuttal period. Otherwise, we hope that the reviewer would consider increasing their score.

---

> ### Author Response · Authors · 2024-11-22
> **Reply to Reviewer qEak**
>
> > While the formula the authors wrote down in the reply is symmetric, the formula in the first half of Eq.(3) is not symmetric wrt to Y and X_j. So, it is confusing to me how the authors arrive at Eq.(3), which is the foundation of the entire method.
>
> We would like to further clarify that the symmetry property of $Y$ and $X_{j}^M$ is **conditional on $X_{j-1}^M$** and in our setting, the first half of Eq. (3) indeed satisfies this symmetry property.
>
> The proof of Proposition 3.4 provided in Appendix A can be used to show this. However, for clarity, we have included explicit and rigorous proof of this statement below.
>
> We prove this as follows:
>
> The first step in our proof involves showing that $p(Y\mid X_j^M) = p(Y\mid X_j^M, X_{j-1}^M)$.
> To see this, recall that $X_{j}^M$ encapsulates the model's CoT generation for all steps up to step $j$. Specifically, $X_{j}^M$ also includes the previous CoT steps $X_{j-1}^M$. Therefore, conditional on $X_{j}^M$, the state $X_{j-1}^M$ is deterministic and hence, $Y \perp\\!\\!\\!\perp X_{j-1}^M \mid X_{j}^M$. From this, it follows that $p(Y\mid X_j^M) = p(Y\mid X_j^M, X_{j-1}^M)$.
>
> Now using this fact, we get that
> $$E[\log p(Y\mid X_j^M)] - E[\log p(Y \mid X_{j-1}^M)]$$
> $$= E[\log p(Y \mid X_j^M, X_{j-1}^M)] - E[\log p(Y \mid X_{j-1}^M)]$$
> Next, the definition of conditional probability yields that the above is equivalent to:
> $$= E\left[\log\frac{ p(Y, X_{j}^M \mid X_{j-1}^M)}{p(X_{j}^M \mid X_{j-1}^M)}\right] - E[\log p(Y \mid X_{j-1}^M)] $$
> $$= E\left[\log\frac{ p(Y, X_{j}^M \mid X_{j-1}^M)}{p(X_{j}^M \mid X_{j-1}^M)\\, p(Y \mid X_{j-1}^M)}\right] \left(=: \mathcal{I}(Y;  X_{j}^M \mid X_{j-1}^M) \right)$$
> Where in the expectation above, both the numerator and denominator (as well as the expectation itself) are symmetric w.r.t. the conditional distributions of $Y$ and $X_{j}^M$conditional on $X_{j-1}^M$.
>
> This proves the equality in Eq. (3) in our paper. Given the symmetry property of the conditional mutual information $\mathcal{I}(Y;  X_{j}^M \mid X_{j-1}^M)$, it then follows that the LHS also follows the symmetry property.
>
> **Possible source of confusion**
>
> There seems to be a slight misunderstanding here, which could be the cause of confusion. Specifically, the symmetry holds **conditional on $X_{j-1}^M$** (and not independent of  $X_{j-1}^M$).
>
> To be rigorous, what this means is that there exists some functional $\mathcal{F}: \mathcal{P} \times \mathcal{P} \times \mathcal{P} \rightarrow \mathbb{R}$ where $\mathcal{P}$ is the space of probability distributions, such that
> 1. the first half of Eq. (3) can be expressed as:
> $$E[\log p(Y\mid X_j^M)]-E[\log p(Y \mid X_{j-1}^M)] = E\left[\mathcal{F}\left(P_{Y\mid X_{j-1}^M}, P_{X_j^M\mid X_{j-1}^M}, P_{Y, X_j^M\mid X_{j-1}^M}\right)\right],$$
> where $P_{Z}$ denotes the distribution of the random variable $Z$
> and
> 2. $\mathcal{F}$ is symmetric w.r.t. its first two arguments, i.e. $\mathcal{F}(p, q, r) = \mathcal{F}(q, p, r) $. Note that there is no symmetry requirement w.r.t. the third argument of $\mathcal{F}$ because the joint distribution $P_{Y, X_j^M\mid X_{j-1}^M}$ is already symmetric w.r.t. $Y$ and $X_j^M$ i.e. $P_{Y, X_j^M\mid X_{j-1}^M}= P_{X_j^M, Y\mid X_{j-1}^M}$.
>
> It can be shown using Proposition 3.4, that the functional $\mathcal{F}$ which satisfies the above conditions is
> $$\mathcal{F}(p, q, r) = - E_{X\sim p}[\log p(X)] - E_{X\sim q}[\log q(X)] + E_{X\sim r}[\log r(X)].$$
> We hope this clarifies the symmetry property of the first half of Eq. (3).

---

> > ### Comment · Reviewer_qEak · 2024-11-24
> >
> > Thanks the authors for the additional clarification. Please include the proof in the updated revision of the paper.

---

### Author Response · Authors · 2024-12-01
**Additional experiments on PRM800K dataset**

To further demonstrate the practical applicability of our method, we have conducted an additional experiment on OpenAI's PRM800k dataset [Lightman et al., 2023] which is obtained by labelling the intermediate steps of the MATH dataset.

More specifically, this dataset is a process supervision dataset containing step-level correctness labels for model-generated solutions to problems from the MATH dataset. To collect this dataset,  Lightman et al., 2023 presented human data-labelers with step-by-step solutions to the MATH problems sampled using a fine-tuned GPT-4 model. The human annotators assigned each step in the solution a label of positive (+1), negative (-1), or neutral (0). A positive label indicates that the step is correct and reasonable. A negative label indicates that the step is either incorrect or unreasonable. A neutral label indicates ambiguity. In practice, a step may be labelled neutral if it is subtly misleading, or if it is a poor suggestion that is technically still valid.

The goal of this experiment was to detect incorrect CoT steps (i.e., the steps labelled as negative by human labellers), using our method as well as the ORM baseline.


**Aggregate results**

The table below shows the aggregate information gain as well as the mean probability of correctness across the positive, negative and neutral sub-steps (along with 2 standard errors):

|  Human annotations  | -1 | 0 | +1 |
| -------- | ------- | ------- | ------- |
| IG (Ours)  |  0.058 ± 0.062  | -0.011 ± 0.050 | 0.168 ± 0.030 |
| ORM |   0.734 ± 0.004  | 0.745 ± 0.008 | 0.744 ± 0.002 |

As expected, these results show that the information gain across incorrect steps (with labels -1) is significantly lower than the information gain for labels +1. Additionally, we observe that the information gain across neutral steps (with labels 0) is also comparatively small, which could be explained by the fact that these steps are deemed irrelevant and hence do not add any useful information regarding the ground truth.

In contrast, the average probability of correctness for the ORM classifier is roughly the same across each label and, on average, is not very informative.

**Sample wise detection**

Additionally, we also used the sample-wise information gain (IG) as well as the ORM baseline to classify if a step is correct (as outlined in Section 3.3.). To avoid ambiguity, we filtered out the neutral sub-steps (with labels 0) for this experiment and considered a balanced held-out dataset with equal number of correct and incorrect steps. The table below shows the sample-wise results for both methods (where we chose the best thresholds for each baseline using a held-out dataset).

|    | Accuracy | TPR $\uparrow$ | FPR $\downarrow$ | TNR $\uparrow$ | FNR $\downarrow$ |
| -------- | ------- | ------- | ------- | ------- | ------- |
| IG (Ours)  |  0.74  | 0.84 | 0.37 | 0.64 | 0.16 |
| ORM |   0.69  | 0.55 |  0.18 |  0.82 | 0.45 |

It can be seen that the accuracy of our method is higher than that of the ORM classifier. Additionally, our method also leads to higher TPR and a lower FNR than the ORM classifier.

The results above show that our method not only remains applicable to more complex datasets such as the MATH data, but additionally, it out-performs the outcome-based baseline in this setting as well.

Finally, we thank the reviewers for their suggestions to improve our paper and will also include these results in the final version of our paper.

**References**

Hunter Lightman, Vineet Kosaraju, Yura Burda, Harri Edwards, Bowen Baker, Teddy Lee, Jan Leike, John Schulman, Ilya Sutskever, Karl Cobbe. Let's Verify Step by Step. 2023

---

### Meta-Review · Area_Chair_upd5 · 2024-12-21

**Metareview:**

Summary:
The paper tries to improve the evaluation of chain-of-thought (CoT) reasoning in large language models (LLMs)  with an information-theoretic approach. The core idea is to measure the "information gain" at each step of reasoning, enabling the detection of failure modes without relying on annotated datasets. Through experiments on toy data and the GSM-8K dataset, the paper demonstrates that this approach identifies errors in CoT more effectively than chosen baseline methods such as outcome-based reward models (ORMs) and Math-Shepherd.

Strengths:
- Interesting solution using information gain
- Provides a mathematical formulation for the framework

Weakness:
- The paper received a mixed response from the reviewers
- Generalizability: Initial experiments were mainly on toy datasets and controlled settings. Questions about applicability to more complex reasoning tasks beyond arithmetic remain. It would be useful to compare to baselines in more natural settings and compare against reported numbers in addition to the newly designed controlled settings.
- Limited Real-World Validation: Only intermediate step accuracy reported, no downstream usage such as verifier or reward model
- Assumptions: The framework makes some strong assumptions about task decomposition and primitive tasks. In particular, the requirement for one-to-one mappings of reasoning steps to primitive tasks is restrictive and may not generalize to real-world tasks.
- Supervisor Model: Still requires training a separate supervisor model, adding computational complexity
- Clarity on Error Types: Distinctions between irrelevant and error steps could be better elucidated within the framework.

Decision:
While the work shows promise and addresses an important problem with an interesting solution, unfortunately, the paper can't be accepted in its current form and addressing all the concerns (e.g. more comprehensive evaluation) would warrant another round of reviewing.

**Additional Comments On Reviewer Discussion:**

We thank the authors and reviewers for engaging during the discussion phase towards improving the paper. Below are some of the highlights:

1. Real-world applicability: Almost all reviewers raised concerns about evaluation
- For complex tasks, authors addressed by adding experiments on PRM800K dataset
- For diverse settings and natural comparison to baselines reported numbers, authors positioned current work as foundational study similar to "Physics of LLMs" papers

2. Supervisor model complexity:
- Authors clarified that while training is needed, it eliminates need for expensive human annotations
- Demonstrated relatively small training data requirements (2000 samples for GSM-8K)

3. Theoretical concerns about symmetry:
- Authors provided detailed mathematical proof of conditional mutual information symmetry
- Reviewer accepted explanation but requested inclusion in the final paper

4. Task decomposition concerns:
- Authors explained framework's flexibility in handling different task types
- Demonstrated application to arithmetic and planning tasks

In summary, though the authors provided some clarifications in rebuttal, many suggested improvements (e.g. more comprehensive evaluation) would require substantial revision beyond what's possible in the rebuttal period.

---

### Decision · Program_Chairs · 2025-01-22

Reject